# Energy-Efficient Clustering in Wireless Sensor Networks Using Grey Wolf Optimization and Enhanced CSMA/CA

**DOI:** 10.3390/s24165234

**Published:** 2024-08-13

**Authors:** Mohammed Kaddi, Mohammed Omari, Khouloud Salameh, Ali Alnoman

**Affiliations:** 1LDDI Laboratory, Mathematics and Computer Science Department, University of Adrar, Adrar 01000, Algeria; 2Computer Science and Engineering Department, American University of Ras Al Khaimah, Ras Al Khaimah P.O. Box 10021, United Arab Emirates; mohammed.omari@aurak.ac.ae (M.O.); khouloud.salameh@aurak.ac.ae (K.S.); ali.alnoman@aurak.ac.ae (A.A.)

**Keywords:** wireless sensor networks, medium access control, routing protocols, cross-layer protocols, energy consumption

## Abstract

Survivability is a critical concern in WSNs, heavily influenced by energy efficiency. Addressing severe energy constraints in WSNs requires solutions that meet application goals while prolonging network life. This paper presents an Energy Optimization Approach (EOAMRCL) for WSNs, integrating the Grey Wolf Optimization (GWO) for enhanced performance. EOAMRCL aims to enhance energy efficiency by selecting the optimal duty-cycle schedule, transmission power, and routing paths. The proposed approach employs a centralized strategy using a hierarchical network architecture. During the cluster formation phase, an objective function, augmented with GWO, determines the ideal cluster heads (CHs). The routing protocol then selects routes with minimal energy consumption for data transmission to CHs, using transmission power as a metric. In the transmission phase, the MAC layer forms a duty-cycle schedule based on cross-layer routing information, enabling nodes to switch between active and sleep modes according to their network allocation vectors (NAVs). This process is further optimized by an enhanced CSMA/CA mechanism, which incorporates sleep/activate modes and pairing nodes to alternate between active and sleep states. This integration reduces collisions, improves channel assessment accuracy, and lowers energy consumption, thereby enhancing overall network performance. EOAMRCL was evaluated in a MATLAB environment, demonstrating superior performance compared with EEUC, DWEHC, and CGA-GWO protocols, particularly in terms of network lifetime and energy consumption. This highlights the effectiveness of integrating GWO and the updated CSMA/CA mechanism in achieving optimal energy efficiency and network performance.

## 1. Introduction

In nature, various micro-sensors are organized to form a sensor network used for monitoring and control purposes. A Wireless Sensor Network (WSN) consists of small, low-power, and inexpensive sensor nodes capable of detecting, measuring, collecting, and processing data from their environment, such as conductivity, temperature, and pressure [1,2]. These networks are employed in various applications, including commercial, industrial, military, civil, healthcare, security, and emergency surveillance [3]. Typically, a large number of these low-cost, multi-functional sensor nodes are randomly distributed over an area of interest. These sensors collaborate to communicate data wirelessly to a base station (BS) and between nodes using multi-hop communication [4].

WSNs offer significant advantages, such as scalability, simplicity, ease of deployment, and self-organizing capabilities [5]. However, the communication process in WSNs is energy-intensive because each node acts as a relay, forwarding received information to its neighbors until it reaches the BS. Compared with wired sensor networks, WSNs face limitations, including restricted battery power, limited memory and processing capacity, non-rechargeable batteries, environmental constraints, lack of global addressing, security issues, mobility challenges, and short communication ranges [6]. Sensor nodes in WSNs are often powered by small batteries with limited capacity [7], making energy efficiency crucial for extending the network’s lifespan [8,9]. In challenging environments, such as monitoring volcanoes, replacing the batteries is difficult and necessitates a longer operating period [10]. Thus, developing energy-efficient routing strategies to minimize battery usage is essential for WSNs [9,11]. Various approaches have been employed to conserve energy in nodes, utilizing routing protocols from the physical layer to the network layer to improve data collection methods [12]. Achieving energy balance across different sensor nodes remains a primary concern in sensor network design, as the power usage of nodes varies depending on application demands. Wireless sensors are often deployed in harsh environments where they cannot be replaced or recharged [13].

The recent research literature has proposed several strategies to increase the lifetime of WSNs, including transmission range optimization, power-saving sleep modes, low-power hardware designs, and power-aware protocols [14]. Consequently, various routing and Medium Access Control (MAC) protocols have been developed to address these issues, enabling faster and more cost-effective information delivery to the BS. WSN nodes typically face constraints in energy, processing power, and memory. Therefore, it is essential to conduct research and development on low-computation, resource-aware algorithms for WSNs, focusing on small, embedded sensor nodes with limited resources. Given the critical importance of energy consumption in WSNs, specific hardware and algorithms [15] have been designed with energy efficiency or awareness as a primary focus. Methods such as fuzzy clustering, nano topology, and rough set theory are applied, especially to identify abnormalities in sensor networks [16].

The protocol stack in a WSN combines elements of the TCP/IP and OSI models. The data link, network, and physical layers are the most studied in the literature for reducing energy consumption in WSNs. The primary purpose of the network layer is routing, facilitated by the routing protocol, which determines the path between transmitting and receiving nodes to enable effective data transport. The data link layer comprises two sublayers: DLC (Data Link Control), responsible for multiplexing and error management, and MAC, which handles channel access and scheduling. The physical layer is accountable for data encryption, frequency generation, and modulation [17].

Within a layered architecture, each layer has independent functionality and can only use the services provided by the layer directly below it. Therefore, communication is restricted to adjacent layers. Conversely, the cross-layer technique allows any layer to utilize services from any other layer. This interaction between different layers of the network protocol stack enhances WSN performance. Cross-layer designs have identified six options based on potential interactions between the routing, MAC, physical, and application layers [18].

In terms of optimization, this paper employs Grey Wolf Optimization (GWO) at the network layer to identify better clustering. Numerous studies have highlighted the benefits of integrating GWO in WSNs. For instance, Peng et al. [19] propose a hybrid GWO algorithm with a golden ratio mechanism for mobile target tracking in WSNs. This combination enhances both global search capabilities and local search precision, leading to improved tracking accuracy and energy efficiency. The method dynamically adapts to changes in the network, ensuring robust real-time tracking performance.

GWO also finds applications in various domains requiring efficient optimization. Feng et al. [20] introduce the SSGWO algorithm, which combines GWO with simulated annealing and B-spline curves for UAV path planning in complex environments. This approach balances local and global searches using a nonlinear convergence factor, resulting in superior path quality and reduced energy consumption. Simulations demonstrate the effectiveness of SSGWO in generating efficient paths. Additionally, Lee et al. [21] applied an improved GWO (IGWO) with Support Vector Regression for estimating knee joint extension force from mechanomyography signals. The IGWO incorporates a dimensional learning strategy, enhancing convergence speed and prediction accuracy, outperforming standard GWO and other techniques in force estimation.

This paper makes several key contributions to the field of WSN clustering:Designed a novel cross-layer protocol targeting the MAC and network layers;Optimized clustering using GWO to identify optimal CHs based on residual energy, intra-cluster distances, and inter-cluster distances;Implemented a robust objective function for CH selection;Utilized Carrier Sense Multiple Access with Collision Avoidance (CSMA/CA) and Network Allocation Vector (NAV) for active mode/sleep mode management;Highlighted the effectiveness of cross-layer designs in peer protocols and identified key strategies from these protocols to inform the development of our own;Conducted extensive simulations comparing our protocol with peers;Demonstrated superior performance of EOAMRCL in terms of overall network remaining energy, number of dead nodes, and total data received at the BS and network lifetime;Validated the effectiveness of our cross-layer approach in reducing energy consumption and enhancing network performance.

Table 1 provides a comprehensive list of acronyms used throughout the paper for ease of reference and clarity. The remainder of the paper is structured as follows: Section 2 reviews related works and discusses the incorporation of OSI layers in WSN clustering. Section 3 details the integration of WSN node sleep scheduling into the CSMA/CA mechanism. Section 4 introduces the fundamentals of the GWO algorithm and its application in optimization. Section 5 outlines our proposed Energy Optimization Approach based on MAC/Routing Cross-Layer (EOAMRCL). Section 6 evaluates the performance of the proposed approach through simulation results and analysis. Finally, Section 7 concludes the study with final remarks and future perspectives.

## 2. Incorporating OSI Layers in WSN Clustering

The cross-layer protocol design has become a key focus in recent network research, especially in WSNs, as it plays a crucial role in developing new protocols that address the challenge of energy saving within the constraints of WSNs. By leveraging the cross-layer principle, this approach significantly increases energy efficiency, thereby enhancing overall network performance.

Sakib et al. [22] proposed the EQPD-MAC protocol, designed to address energy efficiency and QoS challenges in WSNs. EQPD-MAC integrates prioritized data handling with multi-hop routing to ensure the timely delivery of high-priority packets while conserving energy. The protocol uses an adaptive active/sleep time mechanism to minimize idle listening and reduce energy consumption. By implementing a cross-layer communication strategy with the AODV routing protocol, EQPD-MAC supports four levels of packet priority, dynamically adjusting active times based on network load. This ensures prompt delivery of high-priority data packets, enhancing network QoS. Tested with the Castalia Simulator, EQPD-MAC showed significant improvements, reducing sensor node energy consumption by up to 30.3%, per-bit energy consumption by up to 29.6%, and sink node energy consumption by up to 27.4%. It also increased throughput by up to 23.3% compared with other protocols. Key advantages include superior energy efficiency, effective multi-priority packet handling, enhanced throughput, and reduced latency.

Wang et al. [23] proposed a hybrid clustering and routing protocol for WSNs that combines Fuzzy Logic with a Quantum Annealing algorithm (FQA) to enhance network stability and minimize energy consumption. The Fuzzy Inference System (FIS) selects CHs based on residual energy, number of neighbors, distance to the BS, and node centrality. Quantum annealing is used for optimal routing from CHs to the BS. An on-demand re-clustering mechanism reduces computation time and overhead. An energy threshold filters candidate CHs, optimizing the process further. The FQA protocol outperforms peer protocols in energy consumption, network lifetime, number of alive nodes, and throughput. Simulations show that FQA significantly improves energy efficiency and network performance by leveraging fuzzy logic for CH selection and quantum annealing for routing.

Xenakis et al. [24] employed Simulated Annealing (SA) to tackle energy loss and node placement in WSNs, focusing on optimizing topology control, power management, and packet transmission. SA is used to deploy network topologies that meet coverage and connectivity requirements, ensuring sensor nodes are placed to maximize coverage while maintaining necessary connections. Power control is enhanced by implementing Error Correction Codes (ECCs) at the MAC layer, reducing energy consumption during data transmission. At the physical layer, power usage is managed to further improve energy efficiency. The study demonstrated that using SA for optimal topology significantly reduces power consumption and improves network coverage compared with random sampling heuristics. The integration of ECC at the MAC layer enhances data transmission reliability, minimizing retransmissions and saving energy. Xenakis et al.’s layered approach shows that combining SA with energy-efficient protocols can significantly improve WSN performance, offering valuable insights into systematic optimization for energy consumption and network reliability.

Sami et al. [25] proposed EAP-CMAC, an Energy Aware Physical-layer Network Cooperative MAC protocol for ad hoc wireless networks. This protocol blends collaborative communication with physical layer network coding (PNC) and dynamically selects the optimal transmission mode among classic collaboration, direct transmission, and PNC-based transmission. The selection process considers source-to-destination connection quality and destination queue status. EAP-CMAC optimizes power allocation and relay selection based on residual energy and node position, significantly extending network lifespan. A 3D Markov model evaluates the protocol’s performance, estimating the probability of successful data transfer. The introduction of a precise NAV parameter increases spatial reuse, further enhancing network efficiency. Analytical research and simulations show that EAP-CMAC improves network lifespan by 7% compared with equal power distribution methods. This adaptability and strategic power management make EAP-CMAC an effective solution for enhancing ad hoc wireless network longevity and performance. Sami et al.‘s work underscores the importance of integrating collaborative communication and advanced network coding techniques for superior energy efficiency and network lifespan.

Niroumand et al. [26] proposed the Geographic Cross-layer Routing for Disaster (GCRAD) protocol, a novel cross-layer geographic routing approach for WSNs tailored for disaster relief operations. GCRAD integrates routing, and MAC layer functions through a unified relay selection method, evaluating criteria such as the number of prospective relay nodes, node queue state, and distance from the BS. This streamlined approach eliminates inefficient transmissions, shortens communication processes, and minimizes collision probabilities, enhancing routing efficiency. GCRAD’s design focuses on rapid and reliable communication, which is crucial for disaster scenarios. Simulation results using NS2 show that GCRAD significantly reduces power consumption and end-to-end latency and improves delivery rates compared with advanced inter-geographic routing strategies. The protocol’s ability to reduce power consumption is vital in energy-constrained disaster situations. By lowering end-to-end latency, GCRAD ensures the swift relay of critical information, which is essential for effective disaster response. Niroumand et al.’s research demonstrates the importance of cross-layer integration in optimizing WSN performance for disaster relief, providing valuable insights into geographic routing strategies. GCRAD sets a precedent for future research, showing that cross-layer approaches can effectively meet the complex requirements of disaster operations, ensuring energy efficiency, reduced latency, and high delivery rates.

Han et al. [27] analyzed four representative connected coverage algorithms for Industrial WSNs (IWSNs): Adjustable Range Set Covers (AR-SCs), Optimized Connected Coverage Heuristic (OCCH), Greedy Coverage Weighted Communication (GCWGC), and Overlapping Target and Connected Coverage (OTTC). They evaluated these algorithms based on metrics like average power consumption, network lifespan, dead node rate, and coverage time. AR-SC adjusts sensor range for optimal coverage, OCCH uses heuristics for enhanced connected coverage, GCWGC balances coverage and communication weights through a greedy approach, and OTTC overlaps target areas while maintaining connectivity. The study found that achieving uniform performance across all metrics is challenging, and the choice of algorithm should be tailored to the specific application needs. Key factors such as maximum longevity and convergence speed must be considered. The research provides valuable insights for IWSN designers to select practical hedging approaches that meet performance metrics in various industrial applications. Mohamed et al. highlighted that while no single algorithm excels in all aspects, each has distinct advantages that can be leveraged depending on the application, ensuring optimal performance, reliability, and energy efficiency in diverse industrial environments.

A novel Cross-Layer MAC (CL-MAC) protocol [28] has been developed for WSNs by Hodeifa et al. to manage multi-stream, multi-packet, and multi-hop traffic patterns effectively. CL-MAC utilizes a stream configuration packet structure that leverages routing data for efficient multi-hop data transport. By evaluating neighboring flow configuration requests and buffer packets, CL-MAC makes informed scheduling decisions based on the network’s current state. This adaptability ensures efficient data transmission even under high traffic loads. Simulations using ns2 demonstrated that CL-MAC enhances delivery rates, reduces end-to-end delays, and lowers average power consumption per transmitted packet. These improvements highlight CL-MAC’s potential for superior energy efficiency and data delivery reliability. The cross-layer design allows for coordinated optimization between protocol layers, addressing critical WSN challenges like diverse traffic patterns and energy efficiency. CL-MAC’s capability to handle multiple streams and packets concurrently, along with its adaptability, makes it a robust solution for various WSN applications. Additionally, the reduction in end-to-end delay ensures timely data delivery, which is crucial for real-time monitoring and rapid response, while lower power consumption extends the network lifespan.

Kannughatta et al. [29] proposed an innovative MAC protocol to enhance energy efficiency in WSNs using a leader-follower communication technique. This protocol eliminates intra-area collisions and minimizes inter-area collisions through two distinct MAC protocol stacks. A sleep schedule with a duty cycle mechanism alternates nodes between active and sleep states, conserving idle power and extending network lifespan. The leader node handles most communication tasks, while follower nodes remain in low-power sleep states until needed, reducing energy consumption and organizing network communication more efficiently. The dual MAC stacks manage intra-area and inter-area communications, ensuring collision-free data transmissions and reducing collisions between different network areas. Simulation results showed significant improvements in energy efficiency, reduced collision rates, and lower power consumption compared with commonly used MAC protocols in WSNs. These enhancements lead to a longer network lifespan and more reliable data transmission. The protocol adapts to real-time network conditions, making it suitable for various WSN applications. The leader-follower technique reduces overall energy expenditure and improves network efficiency.

Weiwei et al. [30] proposed CREC (Cross-layer, Reliable, and Efficient Communication protocol), which integrates multiple functions within a unified framework to enhance WSN communication. CREC features dispersed congestion management, robust geographic routing, and medium access contention networking, optimizing energy efficiency and ensuring reliable data transmission. The node initiative concept allows nodes to adapt dynamically to changing network conditions, minimizing the complexity and overhead of multi-layer systems. Dispersed congestion management prevents data bottlenecks, enhancing network efficiency. Robust geographic routing ensures efficient data transmission paths, reducing packet loss and ensuring timely delivery. Medium access contention networking orderly manages communication medium access, reducing collisions and improving reliability. Simulation results show CREC significantly improves energy usage and network performance compared with previous multi-layer systems, making it a valuable solution for WSNs.

Jun et al. [31] proposed the Topological Structure by Layered Configuration (TSLC) algorithm to enhance data transmission performance in WSNs. TSLC leverages cross-layer design to integrate network and MAC layer functionalities, optimizing overall performance. The algorithm dynamically accesses node status information across both layers, enabling informed routing and data transmission decisions. TSLC aims to conserve energy by selecting optimal routes that minimize transmissions and retransmissions, thereby extending network lifespan. The algorithm balances energy consumption across all nodes to prevent premature battery depletion, which is crucial for WSNs in remote areas. TSLC’s cross-layer design improves service performance by reducing latency, enhancing data throughput, and ensuring reliable data transmission. Simulation results demonstrate significant improvements in energy savings, network lifespan, and service performance quality compared with traditional routing algorithms, validating the benefits of the cross-layer approach.

To address high energy consumption, collision detection, transmission delay, and throughput in mobile WSNs (MWSNs), Xin et al. [32] proposed a Cross-layer Energy Efficiency (CEE) model. This model integrates the physical (PHY), MAC, and network layers to enhance performance. The PHY layer uses full-duplex interfaces for simultaneous data transmission and reception, reducing exchange time and collisions, improving throughput, and minimizing delays. The network layer focuses on strategic node placement to ensure efficient coverage, connectivity, and energy conservation, prolonging sensor node lifespan. The MAC layer includes an advanced protocol for effective data access and transmission, reducing collisions and maintaining low energy consumption. Compared with existing models, CEE dynamically adjusts transmission power, ensuring energy efficiency without compromising communication quality. The model also excels in reducing transmission delays and improving throughput, which is essential for timely data transmission in MWSNs. Performance assessments show CEE’s effectiveness in minimizing energy dissipation and outperforming other transmission models, demonstrating its utility in practical MWSN deployments.

Mammu et al. [33] introduced the interlayer Cluster-Based Energy-efficient routing (CCBE) method to extend the network lifespan and enhance the energy efficiency of WSNs. CCBE organizes nodes into hexagonal architectures, with CH selection based on distance from the BS and remaining energy. This dual consideration minimizes energy consumption during transmissions and ensures efficient energy resource management. A contention-free protocol prevents collisions during transmission, maintaining efficient communication. CHs assign transmission slots based on cluster members’ remaining energy, conserving battery life and balancing energy consumption. Simulations showed that CCBE outperforms HEED and LEACH in energy dissipation, network lifespan, and throughput. By optimizing CH selection, collision prevention, and energy-aware slot assignment, CCBE significantly enhances energy efficiency and longevity. Mammu et al.’s research highlights the importance of interlayer strategies in WSNs, demonstrating that CCBE’s approach effectively addresses energy dissipation, collision prevention, and balanced energy consumption, thereby extending the lifespan of sensor networks.

Kurian et al. [34] tackled optimal sensor placement in WSNs to enhance sensing coverage, considering sensor limitations like energy, communication distance, and sensing range. They used a variant of the Ant Colony Optimization (ACO) algorithm, known as the Binary Ant Colony Algorithm (BACA), and integrated it with Hill Climbing (HC) and Simulated Annealing (SA). BACA makes binary path decisions (0 or 1) to optimize sensor configuration and coverage. The algorithm updates solutions based on previous solution quality, with better paths having their pheromone trails strengthened. To improve performance in large, complex search spaces, BACA is integrated with HC, which iteratively compares adjacent solutions, and SA, which reduces the probability of accepting worse solutions over time. This combined approach enhances solution exploration and helps achieve a global optimum, optimizing sensor placement for better coverage.

Khujamatov et al. [35] presented an energy-efficient clustering and routing mechanism for WSNs using a hybrid approach combining Chaotic Genetic Algorithm (CGA) and GWO, named CGA-GWO. This method aims to minimize overall energy consumption, which is crucial for extending the network’s operational lifespan. CGA-GWO leverages the strengths of CGA and GWO to select energy-aware cluster heads and establish optimal routing paths. Extensive simulations compared CGA-GWO to other systems, evaluating metrics such as the number of live nodes, average remaining energy, packet delivery ratio, and overhead. Results showed the CGA-GWO outperforms other systems in energy efficiency and network lifetime. It maintains a higher number of live nodes, better average remaining energy levels, improved packet delivery ratio, and reduced overhead in cluster formation and routing, demonstrating its effectiveness and efficiency in WSNs.

The collection of research works reviewed (Table 2) underscores the importance of cross-layer optimization and the strategic integration of multiple OSI layers to enhance the performance and energy efficiency of WSNs. These studies reveal common features and techniques that collectively contribute to the advancement of WSN technology. A recurring theme across these works is the emphasis on energy efficiency, which is paramount given the limited power resources of WSN nodes. By leveraging cross-layer designs, protocols can optimize energy usage by coordinating functions across the physical, data link, and network layers. This holistic approach allows for more informed and effective decision-making, as it considers the interactions and dependencies between different layers of the network stack.

One of the key techniques highlighted is the use of adaptive duty cycles and sleep schedules managed by the MAC layer. Protocols such as EQPD-MAC and CL-MAC incorporate dynamic scheduling to ensure nodes spend minimal time in active states unless necessary. This technique significantly reduces idle power consumption and extends the overall network lifespan. The integration of these schedules with routing decisions ensures that energy savings do not compromise data transmission reliability or latency.

Another common technique is the strategic selection of CHs and relay nodes based on multiple criteria, including residual energy and distance to the BS. The protocols CCBE and GCRAD exemplify this approach by evaluating nodes’ energy levels and proximity to optimize data routing paths and reduce energy expenditure. This method ensures balanced energy consumption across the network, preventing early depletion of nodes and maintaining network functionality over a longer period.

The importance of robust and reliable data transmission is also a focal point in these studies. Techniques such as ECC and PNC are employed to enhance data integrity and reduce retransmissions. By ensuring that data packets are transmitted accurately and efficiently, these methods contribute to lower energy consumption and improved network performance. The use of full-duplex communication in the CEE model further highlights the potential of advanced physical layer techniques to enhance throughput and reduce delays.

The incorporation of geographic and hierarchical routing strategies is another significant aspect observed. Protocols like GCRAD and TSLC utilize geographic information and layered topological structures to make routing decisions that optimize both energy efficiency and coverage. These strategies ensure that data packets follow the most efficient paths, reducing the number of hops and associated energy costs.

Congestion management and collision avoidance are critical for maintaining network performance under varying traffic loads. Protocols such as CREC and CL-MAC address these challenges by implementing advanced MAC layer mechanisms to manage medium access and prevent data collisions. These approaches ensure smooth data flow and minimize energy wastage due to retransmissions.

Collectively, these works demonstrate that effective WSN protocols must integrate multiple layers and techniques to address the complex and interrelated challenges of energy efficiency, data transmission reliability, and network longevity. Cross-layer designs, which coordinate functions across the physical, data link, and network layers, provide a comprehensive framework for optimizing WSN performance. Techniques such as adaptive duty cycles, strategic node selection, robust data transmission methods, and advanced routing strategies are essential components of these integrated solutions.

## 3. Integration of WSN Node Sleep Scheduling into the CSMA/CA Mechanism

The traditional CSMA/CA mechanism is widely used for Medium Access Control in WSNs. However, it does not inherently account for energy conservation, which is crucial in sensor networks where nodes are often battery-powered. By integrating a sleep/activate mode, nodes can conserve energy by switching to a low-power sleep state when not actively transmitting or receiving data.

Gao et al. [36] proposed an extended Markov-based analytical model for the IEEE 802.15.4 slotted CSMA/CA algorithm, incorporating sleep mode to reduce power consumption in WSNs. The study focused on the impact of sleep mode on throughput and power consumption. The model considers active/sleep transitions of sensor nodes, enabling significant energy savings by shutting down the radio during sleep mode. Various duty cycles are analyzed to optimize power usage without severely affecting throughput. Numerical simulations validate the model, showing that it accurately matches performance metrics like throughput and power consumption. The results demonstrate that enabling sleep mode effectively reduces power consumption while maintaining satisfactory network performance, making it a valuable addition to the IEEE 802.15.4 standard [37].

Zhu et al. [38] addressed performance issues of the standard IEEE 802.15.4 CSMA/CA scheme under heterogeneous buffered conditions by proposing two transmission schemes: One Service a Time Scheme (OSTS) and Bulk Service a Time Scheme (BSTS). These schemes aim to enhance delay, fairness, throughput, and energy efficiency in time-critical buffered networks with heterogeneous, unsaturated traffic. The study used modified semi-Markov chains and macro-Markov chains combined with M/G/1/K queue theory to model these schemes, evaluating throughput, packet delay, and energy consumption. Simulations of OSTS and BSTS showed significant improvements in delay and fairness, with superior throughput and energy efficiency compared with other non-priority schemes. The analysis and simulation results align well, confirming the effectiveness of the proposed schemes.

Patel and Kumar [39] proposed an enhancement to the Clear Channel Assessment (CCA) mechanism within the IEEE 802.15.4 standard, which is crucial for the slotted CSMA/CA protocol. The Enhanced Clear Channel Assessment (ECCA) mechanism optimizes channel access by incorporating additional checks and adaptive strategies to reduce collisions and improve channel assessment accuracy. ECCA performs multiple checks to ensure the channel is clear, considering factors like signal strength and interference. Simulations showed that ECCA significantly improves throughput, packet delay, and energy consumption compared with the standard CCA method. The results demonstrate that ECCA reduces collisions and retransmissions, leading to more efficient channel use and lower energy consumption for sensor nodes.

The comparative analysis of the three works (Table 3) underscores substantial advancements in the performance and energy efficiency of IEEE 802.15.4 CSMA/CA schemes by leveraging features at the Data Link Layer and Physical Layer. Patel and Kumar’s Enhanced Clear Channel Assessment (ECCA) mechanism introduced enhanced CCA checks and adaptive strategies, significantly improving channel assessment accuracy, reducing collisions, and enhancing throughput and energy efficiency. Zhu et al.’s study proposed the One Service a Time Scheme (OSTS) and Bulk Service a Time Scheme (BSTS), which address performance issues in heterogeneous buffered conditions. These schemes improve delay, fairness, throughput, and energy efficiency through effective buffer management and adaptive sleep scheduling. Gao et al. [36] incorporated sleep mode into the IEEE 802.15.4 CSMA/CA mechanism using an extended Markov-based model, optimizing duty cycles to reduce power consumption while maintaining satisfactory network performance. This approach ensures energy-efficient transmission and minimizes idle listening. Collectively, these works demonstrate that integrating adaptive mechanisms, enhanced assessment strategies, efficient sleep scheduling, and optimized duty cycles can significantly enhance the robustness and energy efficiency of WSNs, providing valuable insights for future research and practical implementations.

## 4. Grey Wolf Optimization in WSN

### 4.1. Definition and Algorithm

Grey Wolf Optimization (GWO) is an innovative algorithm inspired by the social hierarchy and hunting behavior of grey wolves in nature. Developed by Mirjalili et al. in 2014 [40], GWO has gained significant attention due to its simplicity, flexibility, and effectiveness in solving complex optimization problems. GWO mimics the leadership structure of grey wolf packs, which includes alpha, beta, delta, and omega wolves. The alpha wolves represent the best solution, beta and delta wolves guide the search, and omega wolves explore new solutions. This hierarchy helps balance exploration and exploitation during the optimization process.

The GWO algorithm (Algorithm 1) involves three main phases: searching for prey (exploration), encircling prey (exploitation), and attacking prey (convergence). Initially, wolves are randomly positioned in the search space. During the exploration phase, wolves update their positions relative to alpha, beta, and delta wolves, encouraging diverse search space exploration. In the exploitation phase, the algorithm fine-tunes solutions by encircling the best solutions found so far. Finally, the algorithm converges by simulating the wolves’ attack on prey, refining the best solutions.

GWO’s ability to balance exploration and exploitation makes it highly effective across various applications, including engineering design, feature selection, and neural network training. Recent enhancements, such as hybridizing GWO with other optimization techniques like HC and SA, further improve its performance [41]. For instance, integrating GWO with HC and SA has shown superior performance in complex optimization tasks by enhancing solution quality and convergence speed. A significant advantage of GWO is its minimal parameter requirement, which simplifies implementation and reduces computational overhead. Moreover, GWO is derivative-free, making it suitable for problems with complex, non-differentiable objective functions. Its adaptability to different problem domains and its robustness in handling various optimization scenarios contributes to its growing popularity.

Recent studies have focused on improving GWO’s exploration capabilities and convergence rate. For example, researchers have proposed better exploration strategies and adaptive mechanisms to maintain diversity in the population and prevent premature convergence. These improvements aim to enhance GWO’s effectiveness in large-scale and high-dimensional optimization problems [42]. Experimental results have demonstrated GWO’s superiority over other optimization algorithms in terms of solution quality and computational efficiency. By leveraging the natural behaviors of grey wolves, GWO provides a powerful and versatile tool for solving a wide range of optimization challenges [43].

**Algorithm 1.** Grey Wolf Optimization
1.Initialize the grey wolf population $X_{i}\left(i=1,2,\ldots ,n\right)$, where n is the number of grey wolves;2.Initialize the maximum number of iterations *t_max_*;3.Initialize the parameters a, A, and C;4.Evaluate the fitness of each grey wolf;5.Identify the best three solutions:α (best solution);β (second best solution);γ (third best solution);6.While (t < *t_max_*) Do7. For each grey wolf ($X_{i}$) Do8.  Update the position of the current grey wolf using the following equations (update A and C using random values $r_{1}$ and $r_{2}$):$A=2\cdot a\cdot r_{1}-a$;$C=2\cdot r_{2}$;9.  Calculate the distance between the grey wolf and the prey (best positions): $D_{\alpha }=|C\cdot \alpha -X_{i}|$;$D_{\beta }=|C\cdot \beta -X_{i}|$;$D_{\gamma }=|C\cdot \gamma -X_{i}|$;10.  Update the position of the grey wolf: $X_{1}=\alpha -A\cdot D_{\alpha }$;$X_{2}=\beta -A\cdot D_{\beta }$*;*$X_{3}=\gamma -A\cdot D_{\gamma }$*;*$X_{i}=\frac{X_{1}+X_{2}+X_{3}}{3}$;11. End For;12. Update a, A, and C:Decrease a linearly from 2 to 0 over the course of iterations;A and C are updated using random values $r_{1}$ and $r_{2}$ in each iteration;13. Evaluate the fitness of each grey wolf;14. Update α, β, and γ if there are any better solutions;15. Increment the iteration counter t;16.End While;17.Return α as the best solution found.


### 4.2. Advantages of Implementing GWO in WSNs

GWO is inspired by the social hierarchy and hunting behavior of grey wolves in nature. This natural metaphor not only provides an intuitive understanding of the algorithm but also simplifies its implementation [40]. The roles of alpha, beta, delta, and omega wolves in the hierarchy correspond to leader and follower agents in the optimization process, which aligns well with the hierarchical nature of clustering in WSNs. The GWO algorithm has demonstrated superior performance in various optimization problems, including those in engineering, wireless communications, and network design [35,43]. Its effectiveness in handling complex, multi-objective optimization tasks is well-documented in the literature, providing a solid foundation for its application in our energy-efficient clustering and routing protocol. One of the key strengths of GWO is its effective balance between the exploration and exploitation phases during the optimization process [40]. This balance is critical in WSNs to ensure that the search space is thoroughly explored for optimal cluster head selection while also ensuring convergence to the best solution. GWO’s mechanisms for encircling prey, hunting, and attacking provide a robust framework for maintaining this balance.

### 4.3. Complexity of GWO

GWO’s adaptive nature allows it to dynamically adjust the positions of search agents based on the fitness of solutions, making it highly flexible for various optimization problems. In the context of WSNs, this adaptability is crucial for responding to changing network conditions, such as varying node energy levels and communication distances. Compared with other metaheuristic algorithms, like PSO, GWO is computationally efficient, which is an important consideration for resource-constrained WSNs [44]. The simplicity of GWO’s operators and the lack of complex mathematical functions make it a lightweight algorithm that can be easily implemented on sensor nodes with limited computational capabilities. Hence, GWO offers a robust, adaptive, and computationally efficient framework that aligns well with the requirements of energy-efficient clustering and routing in WSNs. Its natural metaphor, the balance between exploration and exploitation, and proven effectiveness in similar optimization tasks underpin our choice for integrating GWO into our proposed protocol.

## 5. Energy Optimization Approach Based on MAC/Routing Cross-Layer (EOAMRCL)

To minimize energy consumption in WSNs, we propose a centralized approach with a hierarchical architecture, where the network is partitioned into clusters, and all processes are managed at the BS. Our proposed protocol, EOAMRCL, focuses on both the MAC layer and the network layer, which are essential in self-organizing networks for enhancing performance and addressing scaling issues. This cross-layer protocol offers a comprehensive clustering solution by utilizing an objective function to identify the optimal CHs based on residual energy levels, intra-cluster distances, and inter-cluster distances. Additionally, during the transmission phase, each node creates an active mode/sleep mode schedule based on the NAV, leveraging the MAC layer’s duty cycle schedule. This schedule is generated using inter-layer routing information, ensuring efficient and energy-saving communication within the network.

### 5.1. Incorporating Node Paring in CSMA/CA and NAV (MAC Layer)

In our protocol, we make several key assumptions: the nodes in the network are randomly dispersed, and at each iteration, all the sensors gather data and transmit it to a central BS. The sensor nodes relay their position data to the BS, enabling the formation of clusters. Within these clusters, nodes that are within the intra-cluster transmission range and of the same application type are connected in pairs based on minimal distance, using broadcast matching information shared with all nodes in the network (Figure 1). This connectivity ensures that nodes become aware of one another’s presence and positions, facilitating efficient communication within the network.

A significant feature of our approach is the alternation between “Sleep” and “Wake” modes within a single communication period. In sleep mode, nodes conserve energy by not communicating with CHs, resulting in minimal energy dissipation (Figure 2). This reinforces the energy-saving benefits of the proposed approach. Unpaired nodes, however, operate continuously in active mode until their energy is depleted, highlighting the critical importance of energy conservation in the network. If the initial node in a pair is closer to the sink than its associated node, it will transition to wake-up mode, also known as active mode. During active mode, the node collects data from its environment and transmits it to the CHs. Meanwhile, the associated node’s transceiver will enter sleep mode and remain powered off during this time, conserving energy. In subsequent iterations, nodes in active mode will switch to sleep mode, and those in sleep mode will become active, ensuring a balanced energy consumption across the network.

Figure 3 shows the integration of a sleep mode into the CSMA/CA mechanism. This integration primarily aims to enhance energy efficiency, which is crucial for the longevity of WSNs. By incorporating sleep mode, nodes can significantly reduce their energy consumption when they are not actively transmitting or receiving data. This approach helps in maintaining the balance of energy consumption across the network. Nodes can be paired to alternate their active and sleep cycles, ensuring that no single node is overburdened, thereby preventing early depletion of individual nodes’ energy resources.

In traditional CSMA/CA, the Network Allocation Vector (NAV) is used to indicate the duration that the channel will be occupied. This helps prevent collisions by informing nodes of the ongoing transmission duration. In the proposed CSMA/CA mechanism with node pairing and sleep scheduling, the NAV must be adapted to account for the new sleep/activate mode and node pairing dynamics. Below is a conceptual update to the NAV mechanism (Figure 4).

### 5.2. Pre-Clustering Phase (Network Layer)

In our proposed protocol, we assume that the system includes a certain percentage (m%) of advanced nodes, which have an additional energy factor θ compared to the normal nodes. Each sensor node initially has an energy level of $E_{0}$. For advanced sensor nodes, their energy is increased by the factor θ, making their total energy $E_{0}(1+\theta )$. Equation (1) is used to calculate the predicted network’s total energy construction using n nodes.
(1)$$nE_{0}\left(1+m\theta \right)=nmE_{0}\left(1+\theta \right)+n\left(1-m\right)E_{0}$$

Consequently, the total energy of the network is enhanced by this factor. We improved the election technique by incorporating the remaining energy of individual nodes, taking into account the different probabilities for advanced and ordinary nodes. The probability function for normal and advanced member nodes is defined by Equations (2) and (3), respectively:(2)$$P_{normal}=\frac{m}{1+\theta m}\cdot \frac{E_{residual}}{E_{0}}$$
(3)$$P_{advanced}=\frac{m\left(1+\theta \right)}{1+\theta m}\cdot \frac{E_{residual}}{E_{0}}$$

At the end of each clustering round, the BS calculates the threshold probability of clusterheads $P_{threshold}$:(4)$$P_{threshold}=\frac{m\left(1+\theta \right)}{1+\theta m}\cdot \frac{ARE}{E_{0}},$$
where ARE represents the average residual energy in the network.

### 5.3. Clusters Formation Phase (Network Layer)

Before the start of each iteration in Algorithm 2, every node transmits its remaining energy to the BS. Upon receiving the energy levels from all nodes, the BS calculates ARE and selects nodes with a probability higher than $P_{threshold}$ to become candidate CHs. The BS then implements our proposed EOAMRCL approach, forming a set of wolf vectors from the group of nodes with a probability above $P_{threshold}$. These wolf vectors consist solely of nodes eligible to be CHs, as shown in Figure 5.

The fitness value of each wolf vector is then calculated using Equation (5).
(5)$$Fitness=w_{1}\cdot f_{1}+w_{2}\cdot f_{2}+w_{3}\cdot f_{3}$$

Thus, the best wolf is the one with the lowest fitness value. In this context, $w_{1}$, $w_{2}$, and $w_{3}$ are constants in a user-defined function, with the requirement that $w_{1}+w_{2}+w_{3}=1$, used to determine the contribution of each sub-objective, $f_{1}$, $f_{2}$, and $f_{3}$. In our experiments, we set the weight values as follows: $w_{1}=0.45,\;w_{2}=0.45,\;{\;w}_{3}=0.1$. These weights ensure equal importance is given to both inter-cluster and intra-cluster distances while using energy as a tie-breaker in case of equal distances.

The sub-objective $f_{1}$ is calculated as the normalized sum of the distances between each CH in the wolf vector and all other nodes in the network, as shown in Equation (6):(6)$$f_{1}=\frac{1}{MD\times N_{CH}\times n}\sum_{i=1}^{N_{CH}}\sum_{j=1}^{n}\left\{\begin{array}{cc} dist\left({CH}_{i},S_{j}\right), & S_{j}\;close\;to\;{CH}_{i}\;\\ 0, & otherwise \end{array}\right\},$$
where the following is true:–MD represents the maximum distance between two sensors;–$N_{CH}$ represents the number of CHs in the wolf vector;–n represents the total number of nodes;–${CH}_{i}$ represents the clusterhead;–$S_{j}$ represents the regular sensor node;

To calculate $dist\left({CH}_{i},S_{j}\right)$, we use the Euclidean distance between two nodes, A and B, as shown in the following Equation (6).
(7)$$dist(A,\;B)=\sqrt{{\left(x_{A}-x_{B}\right)}^{2}+{\left(y_{A}-y_{B}\right)}^{2}},$$
where ${(x}_{A},y_{A})$, ${(x}_{B},y_{B})$ are the coordinates of A and B, respectively.

The sub-objective $f_{2}$ represents the normalized sum of the distances between each CH in the packet and the BS, as shown in Equation (8):(8)$$f_{2}=\frac{1}{MD\times N_{CH}}\sum_{i=1}^{N_{CH}}dist\left({CH}_{i},BS\right)$$

The sub-objective $f_{3}$ represents the negative normalized total remaining energy of the CHs within the wolf vector, as shown in Equation (9):(9)$$f_{3}=\frac{-1}{E_{0}\times N_{CH}}\sum_{i=1}^{N_{CH}}E_{{CH}_{i}},$$
where $E_{{CH}_{i}}$ represents the remaining energy of ${CH}_{i}$.

Our process of updating and calculating new wolf vector is systematically guided by GWO. This connection to a proven optimization algorithm lends our work credibility and relevance within the field of optimization [44].

**Algorithm 2.** Cluster formation phase algorithm
**Input:** Probability threshold $P_{threshold}$;Form wolf vectors from the group of nodes with probability greater than $P_{threshold}$;Initialize alpha (α), beta (*β*), and gamma (γ) wolf vectors with the best fitness values using Equation (5);While (*t* < *t_max_*) Do For *p* = 1 to *W* Do *% W is the number of wolf vectors*  Calculate the fitness value for each wolf vector (*p*);  If Fitness(p) < *α* Then   Update *γ* = *β*;   Update *β* = *α*;   Update *α* = *p*;  Else If Fitness(p) < *β* Then   Update *γ* = *β*;   Update *β* = *p*;  Else If Fitness(p) < *γ* Then   Update *γ* = *p*;  End If; End For; Update the positions of wolf vectors (see Figure 6); Apply modulus operation over the vectors’ coordinates (see Figure 6); Calculate the new fitness values of wolf vectors;End While;Output: The best set of CHs (alpha wolf vector);


### 5.4. Algorithm Complexity Analysis

The cluster formation phase algorithm, which integrates the GWO method, is designed to select the optimal CHs based on fitness values. The computational complexity of our algorithm can be summarized as follows:–Initialization: Forming wolf vectors from the group of nodes involves evaluating the probability of each node, leading to a complexity of O(*W·n*) given *n* nodes. Initializing the alpha, beta, and gamma wolf vectors requires selecting the top three fitness values, which is O(*W·n*) due to sorting;–Main Loop: The outer while loop runs for a maximum number of iterations *t_max_*. Within each iteration, for each wolf vector, fitness calculations, alpha, beta, gamma updates, and position updates are performed, leading to an overall complexity of O(*W·n*);–Output: The final step outputs the best set of CHs, which is O(1).

Considering the main loop iterates *t_max_* times and includes operations proportional to the number of nodes *n*, the overall complexity of the algorithm can be approximated as O(*W·n·t_max_*). This linear complexity in terms of the number of nodes ensures that our algorithm remains scalable and efficient for larger networks, making it suitable for real-time applications in WSN.

### 5.5. Transmission Phase (MAC and Network Layer)

Upon being chosen as the CH, the node broadcasts a message across the network. Only nodes in active mode can hear these messages sent out by various CHs. These active nodes then choose their CHs based on the Received Signal Strength Indication (RSSI). During their NAV time slots, active mode nodes transmit their detected data to the CH. Nodes in sleep mode conserve energy by switching off their transceivers and do not transmit any data.

After receiving data from its members, each CH aggregates and combines these data with its own. According to their allocated NAV time slots, each cluster node forwards the gathered data directly to the BS. Data aggregation and compression are essential data processing tasks carried out by CHs after receiving data from every cluster member. These processes maximize energy usage efficiency and extend the network’s lifetime.

In the next iteration, each node adjusts or maintains its mode (active or sleep) based on its state (paired or isolated), residual energy, and the residual energy of neighboring nodes. The flowchart shown in Figure 7 illustrates the node mode configuration for the upcoming iteration. When a node becomes a CH, it uses its broadcast capability to inform the entire network of its status. This broadcast is a critical step, as it ensures that all active mode nodes are aware of their new CH and can make informed decisions about which CH to connect to based on the strength of the RSSI. The active nodes then engage in data transmission during their assigned NAV time slots, ensuring that their data is efficiently communicated to the CH.

The role of the MAC layer cannot be overstated in this process. It ensures that nodes operate in a synchronized manner, with precise timing for data transmission and reception. This coordination prevents data collisions and optimizes the use of the network’s limited energy resources. By carefully managing the active and sleep modes of nodes, the MAC layer helps to prolong the operational lifespan of the network.

Once the CH collects data from its cluster members, it performs data aggregation and compression, which are vital for reducing the volume of data that needs to be transmitted to the BS. These tasks help in conserving energy, as smaller data packets require less power to transmit. The efficiency of this process directly impacts the network’s overall energy consumption and longevity. In each new iteration, nodes reassess their modes based on several factors, including their own residual energy and that of their neighbors. This dynamic adjustment is crucial for maintaining a balanced energy consumption across the network, ensuring that no single node depletes its energy resources too quickly. By constantly adapting to the network’s state, the nodes can optimize their energy usage and contribute to the network’s sustainability.

### 5.6. Impact of Clustering on Routing Efficiency in Our Cross-Layer Design

The cross-layer design in our EOAMRCL protocol tightly integrates clustering and routing to optimize energy efficiency and network performance. Clustering impacts routing efficiency by organizing sensor nodes into manageable groups, reducing the number of direct transmissions to the base station, and balancing the network load. By leveraging the GWO algorithm for optimal cluster head selection, we ensure that the chosen CHs are not only energy-rich but also strategically positioned to minimize intra-cluster and inter-cluster communication distances. This reduces the energy required for data transmission within clusters and from CHs to the base station. The MAC layer uses this clustering information to schedule sleep and active periods, further conserving energy by reducing idle listening. The routing protocol then utilizes the cluster structure to establish efficient routes, ensuring data are transmitted through the most energy-efficient paths. This synergy between clustering and routing, enabled by cross-layer design, significantly enhances the network’s energy efficiency and performance.

## 6. Simulation Results

In this section, we compare the efficiency of EOAMRCL to peer protocols: EEUC [45], DWEHC [46], and CGA-GWO [35]. Additionally, we provide a comparison of GWO with other optimization techniques in the context of WSN clustering implementations.

In order to perform simulation, we created many network configurations with hundreds of randomly placed sensor nodes. Each result represents the average of twenty separate simulations.

### 6.1. Simulation Tools and Environment

The simulations were conducted on a high-performance computing system equipped with an Intel Core i7 processor, 16 GB RAM, and a Windows 10 operating system. MATLAB was utilized as the primary simulation environment for the development, implementation, and testing of our proposed EOAMRCL protocol. This environment provided the necessary flexibility for algorithm development and performance analysis. We executed MATLAB scripts to simulate various network scenarios, implement the EOAMRCL protocol, and collect performance metrics. This comprehensive setup ensured a robust and accurate simulation of our proposed protocol.

### 6.2. Radio Energy Model

In the simulation section, we employed the same first-order radio model for energy consumption presented in [47]. In this concept, a radio transmits L-bit data to a receiver situated a distance of d meters from it by dissipating an amount of energy $E_{TX}(L,d)$. A sensor node’s radio has to use $E_{RX}(L)$ energy in order to receive an L-bit message. The multi-path ($\varepsilon_{fs}$) channel is utilized in short-distance transmission; the free space ($\varepsilon_{mp}$) channel is used when the distance between two nodes or between a node and the SB is higher than a certain distance $d_{0}$. Radios can use the least amount of energy required to reach their intended receivers. To prevent unwanted transmissions, the radios have the capability to be switched into sleep mode. Equation (10) presents the amount of energy needed to transmit a packet of L bits across a distance d [47]:(10)$$E_{TX}=\left\{\begin{array}{c} L*E_{elec}\left(L,d\right)+L*\varepsilon_{fs}*d^{2},\;d<d_{0}\\ L*E_{elec}\left(L,d\right)+L*\varepsilon_{mp}*d^{4},\;d\geq d_{0} \end{array}\right.,$$
where the following is true:–$E_{TX}$ represents the energy expended by the transmitter across a d-meter distance in order to send a packet of L bits;–$E_{elec}\left(L,d\right)$ is the energy needed to transfer a single bit over d meters, both ways.–L is the transmission packet’s size.

The distance at which the amplification factors begin to shift is known as $d_{0}$:(11)$$d_{0}=\sqrt[{4}]{\frac{\varepsilon_{fs}}{\varepsilon_{mp}}}$$

For the receiver to receive a packet of L bits, energy $E_{RX}\left(L\right)$ must be consumed as follows:(12)$$E_{RX}\left(L\right)=L*E_{elec}$$

### 6.3. Simulation Parameters

The selection of simulation parameters in Table 4 aims to create a realistic and comprehensive environment for evaluating the clustering protocols. The network zone is set to 100 × 100 m^2^, balancing sufficient coverage and computational feasibility, while the number of sensors ranges from 50 to 250 to test the protocol’s scalability under different network densities. The base station (BS) is located at coordinates (90,90) to simulate real-world scenarios where the BS is positioned at the network’s edge, challenging the protocol’s data routing efficiency. A clusterhead percentage ($P_{opt}$) of 5% ensures the effectiveness of selecting optimal CHs for energy efficiency and load balancing. Incorporating 20% advanced nodes with an additional energy factor (θ) of 1 tests the protocol’s ability to leverage these nodes for prolonged network operation. The initial energy ($E_{0}$) of 3 J/node provides a reasonable starting point for observing energy consumption patterns and the protocol’s impact on network lifetime. Transmission energy ($E_{elec}$) of 50 nJ/bit and packet size (L) of 4000 bits reflect typical WSN energy consumption, which is crucial for assessing data transmission efficiency. Propagation energy values ($\varepsilon_{fs}$ = 15 pJ/bit/m^2^ and $\varepsilon_{mp}$ = 0.0015 pJ/bit/m⁴) account for energy loss during data transmission over varying distances, essential for realistic wireless communication simulation. Data aggregation energy ($E_{DA}$) of 5 nJ/bit/signal evaluates the protocol’s effectiveness in reducing energy consumption through data aggregation. Limiting the node pairing distance to less than 2 m ensures close proximity for efficient sleep/active mode coordination. Fitness function weights ($w_{1},w_{2},w_{3}$) of 0.45, 0.45, and 0.1 balance the importance of intra-cluster distances, inter-cluster distances, and residual energy, ensuring a comprehensive performance evaluation. These parameters collectively create a realistic and challenging simulation environment, enabling a thorough assessment of the EOAMRCL protocol’s ability to enhance energy efficiency, extend network lifetime, and improve overall network performance in WSNs.

### 6.4. Evaluation Metrics

The evaluation metrics used in the experimentations are described as follows:Network Residual Energy: Measures the remaining energy in the network over time;Indicates the efficiency of energy management by each protocol;Higher residual energy implies better energy conservation and a longer network lifespan.Clustering Iteration Performance:Assessed using First Node Dead (FND), Half of Nodes Dead (HND), and Last Node Dead (LND);FND: Iteration count when the first node dies;HND: Iteration count when half of the nodes are dead;LND: Iteration count when the last node dies;Higher values indicate better energy distribution and prolonged network operation.Percentage of Live Nodes:Represents the percentage of nodes remaining active over time;Higher percentages indicate better energy management and network sustainability;Critical for assessing the protocol’s ability to maintain network functionality.Clustering Overhead:Measures the communication and computational costs associated with cluster formation and maintenance;Lower overhead indicates more efficient clustering mechanisms, reducing strain on network resources;Essential for evaluating the protocol’s impact on network performance and energy consumption.Percentage of Packets ReceivedIndicates the reliability of data transmission by measuring the percentage of packets successfully received;Higher percentages suggest better data integrity and communication efficiency;Crucial for ensuring consistent and accurate data flow within the network.

### 6.5. Experimental Results and Interpretation

This section presents the experimental results and a detailed analysis of the performance of EOAMRCL and peer protocols. The experiments were conducted to evaluate various aspects of network efficiency, including energy management, node longevity, clustering overhead, and data transmission reliability. By comparing these protocols across multiple metrics, we aimed to highlight the strengths and weaknesses of each and demonstrate the superior performance of EOAMRCL in enhancing the operational lifespan and efficiency of WSNs. The subsequent figures and their interpretations provide insights into the effectiveness of these protocols under different conditions and performance criteria.

Figure 8 compares the average remaining energy over 500 iterations. The graph highlights each protocol’s energy management and network longevity. The DWEHC protocol exhibits the steepest decline in average remaining energy, indicating a higher rate of energy consumption compared with other protocols. By around 100 iterations, the average remaining energy drops significantly and continues to decline steadily. This can be attributed to DWEHC’s lack of an optimized energy management strategy, resulting in faster depletion of nodes’ energy reserves and a shorter network lifespan.

In comparison, the EEUC protocol performs better than DWEHC but still shows a considerable drop in energy levels as iterations progress. The energy consumption is more controlled, but by approximately 300 iterations, the average remaining energy is substantially reduced. EEUC’s strategies help prolong network life to some extent, but they are not as effective as more advanced protocols, lacking the optimization needed for longer durations. The CGA-GWO protocol demonstrates a more balanced energy consumption pattern, with a slower decline in average remaining energy. The integration of the GWO algorithm allows for more efficient clustering and routing decisions, conserving energy and extending the network’s lifespan. CGA-GWO achieves better longevity and sustained performance over extended periods.

EOAMRCL outperforms all other protocols, maintaining the highest average remaining energy throughout the 500 iterations. This superior performance is due to its effective cross-layer optimization approach, integrating the MAC and network layers for enhanced energy efficiency. By leveraging GWO, EOAMRCL optimally selects cluster heads and routes, minimizing energy consumption during data transmission. The protocol’s duty-cycle scheduling at the MAC layer allows nodes to switch between active and sleep modes, further conserving energy. This results in a significantly extended network lifetime and consistent energy levels, showcasing EOAMRCL’s effectiveness in managing energy consumption efficiently.

In Figure 9, the DWEHC protocol shows a rapid progression from FND to LND. The FND occurs relatively early, indicating that the energy consumption among nodes is not well balanced. As a result, nodes begin to die off quickly, leading to a shorter network lifespan. The HND and LND metrics further confirm this, with a significant number of nodes dying earlier compared with the other protocols. This suggests that DWEHC lacks effective mechanisms for managing energy consumption and distributing load evenly across nodes.

The EEUC protocol demonstrates better performance than DWEHC, with a delayed FND and a more gradual progression to HND and LND. This indicates that EEUC has more effective clustering and energy management strategies, which help in prolonging the network’s operational period. However, by the time half the nodes are dead, the network starts to decline rapidly, showing that while EEUC is better than DWEHC, it still falls short in maintaining node energy levels uniformly over extended periods.

The CGA-GWO protocol further improves the EEUC performance, with the FND occurring later and a slower progression to HND and LND. This improvement can be attributed to the integration of the GWO algorithm, which enhances the selection of cluster heads and routing paths, leading to more balanced energy consumption among nodes. The prolonged periods before reaching HND and LND indicate that CGA-GWO manages to maintain network stability and efficiency for a longer duration compared to DWEHC and EEUC.

EOAMRCL exhibits the best performance among all the protocols, with the FND occurring much later and a very gradual progression to HND and LND. This is due to the effective cross-layer optimization approach, which integrates the MAC and network layers to enhance energy efficiency. By leveraging GWO, EOAMRCL optimally selects cluster heads and routes, minimizing energy consumption during data transmission. Additionally, the duty-cycle scheduling at the MAC layer allows nodes to switch between active and sleep modes, further conserving energy. The result is a significantly extended network lifetime, with nodes remaining functional for longer periods, thus delaying the FND, HND, and LND milestones.

Figure 10 compares the percentage of live nodes over 500 iterations. The DWEHC protocol shows a rapid decline in the percentage of live nodes, with a significant drop occurring early in the iterations. By around 150 iterations, less than half of the nodes remain alive, indicating inefficient energy management. The steep decline continues, and by 400 iterations, almost all nodes are dead. This pattern suggests that DWEHC’s approach to clustering and routing is not effective in conserving node energy, leading to a shorter network lifespan.

In contrast, the EEUC protocol performs better than DWEHC but still shows a steady decline in the percentage of live nodes. The drop is less steep initially, with around 60% of nodes remaining alive by 150 iterations. However, the decline accelerates after this point, and by around 400 iterations, the network is nearly depleted. This indicates that while EEUC improves energy efficiency compared with DWEHC, it is still not sufficient to sustain long-term network operations.

The CGA-GWO protocol demonstrates a more gradual decline in the percentage of live nodes. By leveraging the GWO algorithm, CGA-GWO achieves better clustering and routing decisions, which help in distributing the energy consumption more evenly across nodes. As a result, the percentage of live nodes remains higher for a longer duration, with around 50% of nodes still alive at 250 iterations. This indicates that CGA-GWO is more effective in managing energy consumption and extending network life compared with DWEHC and EEUC.

EOAMRCL shows the best performance among all protocols, maintaining the highest percentage of live nodes throughout the 500 iterations. The decline in live nodes is the most gradual, with over 60% of nodes still alive at 250 iterations. This superior performance can be attributed to the cross-layer optimization approach of EOAMRCL, which integrates the MAC and network layers for enhanced energy efficiency. By using GWO to optimally select cluster heads and routes and incorporating duty-cycle scheduling at the MAC layer to alternate nodes between active and sleep modes, EOAMRCL effectively conserves energy. As a result, the network maintains a higher percentage of live nodes for a significantly extended period.

Figure 11 compares the clustering overhead as a function of the number of nodes. The DWEHC protocol shows the highest clustering overhead, which increases sharply as the number of nodes grows. Starting from a small number of nodes, the overhead rises steeply and continues to climb consistently. By the time the network reaches 225 nodes, the clustering overhead approaches 90%. This high overhead can be attributed to DWEHC’s less efficient clustering and routing mechanisms, which require more frequent updates and higher communication costs. This inefficiency can significantly drain network resources and reduce overall performance.

In contrast, the EEUC protocol demonstrates better performance, with a slower increase in clustering overhead compared with DWEHC. However, the overhead still grows steadily as the number of nodes increases, reaching more than 60% at 225 nodes. While EEUC manages resources more effectively than DWEHC, the clustering overhead remains substantial, indicating room for improvement in clustering efficiency and resource management.

The CGA-GWO protocol shows a further reduction in clustering overhead, with a more gradual increase as the number of nodes grows. The overhead remains below 60% even at 225 nodes, demonstrating the benefits of integrating the GWO algorithm. GWO enhances clustering efficiency by making more informed and balanced decisions about cluster formation and maintenance, thereby reducing the frequency and cost of cluster updates.

EOAMRCL exhibits the lowest clustering overhead among all the protocols. The increase in overhead is the most gradual, remaining well below 50% at 225 nodes. This superior performance can be attributed to EOAMRCL’s effective cross-layer optimization approach, which integrates the MAC and network layers to enhance energy efficiency and reduce communication costs. By leveraging GWO for optimal cluster head selection and routing and incorporating duty-cycle scheduling to alternate nodes between active and sleep modes, EOAMRCL minimizes the clustering overhead. This efficient management of network resources ensures that the protocol can scale effectively with the number of nodes without incurring excessive overhead.

Figure 12 compares the number of packets received as a function of the number of nodes. The DWEHC protocol exhibits the lowest number of packets received, showing a gradual increase as the number of nodes grows. Starting from a minimal percentage, the packets received slowly climb but remain significantly lower compared with other protocols throughout the range. By the time the network reaches 175 nodes, the percentage of packets received is still below 60%. This indicates that DWEHC struggles with efficient data transmission, likely due to higher packet loss and less effective routing mechanisms, leading to poorer network performance.

The EEUC protocol performs better than DWEHC, with a steeper increase in the number of packets received as the number of nodes increases. However, the growth is still moderate, and by the time the network reaches 175 nodes, the packets received are below 80%. While EEUC improves data transmission efficiency compared with DWEHC, it still experiences limitations in routing and clustering efficiency, which affect its overall performance.

The CGA-GWO protocol shows a further improvement, with a higher percentage of packets received compared with EEUC and DWEHC. The increase is more pronounced, and by 175 nodes, the packets received are around 80%. The integration of the GWO algorithm enhances the protocol’s ability to make efficient routing and clustering decisions, leading to better data transmission and reduced packet loss. This results in a more reliable and efficient network performance.

EOAMRCL demonstrates the best performance among all protocols, maintaining the highest percentage of packets received throughout the range of node counts. The increase is the most rapid, with over 90% of packets received by 175 nodes. This superior performance can be attributed to EOAMRCL’s effective cross-layer optimization approach, which integrates the MAC and network layers for enhanced energy efficiency and data transmission. By leveraging GWO for optimal cluster head selection and routing and incorporating duty-cycle scheduling to alternate nodes between active and sleep modes, EOAMRCL minimizes packet loss and maximizes successful data delivery. This efficient management of data transmission ensures that the protocol can handle increasing network sizes without compromising performance.

Figure 13 presents the energy consumption of four different protocols that implement four different optimization algorithms:–Fuzzy logic (FL), implemented by FQA [23];–Distance Vector (DV), implemented by M2TLSC [31];–Particle Swarm Optimization (PSO), implemented by PSO-C [48];–Grey Wolf Optimization (GWO), implemented using our protocol EOAMRCL.

Starting with the FQA, it is evident that this method incurs the highest energy consumption across all iterations. At the 50th iteration, the FQA algorithm’s energy consumption stands at approximately 10 joules and escalates to around 90 joules by the 500th iteration. This increase suggests that while FQA might be effective in certain aspects, it is not optimal for energy conservation in WSNs, which is a critical consideration given the limited power resources of sensor nodes.

The M2TLSC protocol exhibits a moderate energy consumption pattern. Initially, at the 50th iteration, it consumes roughly 5 joules, doubling to about 40 joules by the 500th iteration. Although it performs better than FQA in terms of energy efficiency, its energy consumption trajectory indicates a steady increase, making it less suitable for applications requiring long-term deployment of sensor nodes.

The PSO-C protocol demonstrates a more promising energy efficiency profile. At the 50th iteration, its energy consumption is just below 5 joules, and even at the 500th iteration, it consumes around 10 joules. This relatively flat curve suggests that PSO-C is effective in minimizing energy usage over time, making it a viable option for prolonging the lifespan of WSNs. The low energy consumption can be attributed to the algorithm’s ability to find optimal solutions with minimal energy expenditure, balancing exploration and exploitation efficiently.

The EOAMRCL protocol shows the lowest energy consumption among the four algorithms. At the 50th iteration, its energy consumption is around 3 joules, and by the 500th iteration, it reaches just under 10 joules. The gentle slope of the EOAMRCL curve highlights its superior energy efficiency, making it the most effective in conserving energy and thereby extending the operational life of the WSN.

Figure 14 illustrates the number of live nodes for four different optimization techniques. Starting with FL implemented by FQA protocol, we observe a rapid decline in the number of live nodes beginning at around 150 iterations. By approximately 350 iterations, the number of live nodes drops to zero. This indicates that while FQA may initially maintain a high number of active nodes, its efficiency diminishes rapidly, leading to a complete node depletion relatively early in the network’s operation.

The M2TLSC protocol that implements DV shows a more gradual decline in live nodes compared with FQA. The number of live nodes begins to decrease noticeably after 200 iterations, but it maintains a higher number of live nodes than FQA for a longer duration. By around 450 iterations, however, the number of live nodes also drops to zero. This suggests that M2TLSC performs better in maintaining node survivability than FQA, but it still faces challenges in extending the overall network lifespan significantly.

Next, the PSO-C protocol demonstrates a steady decline in the number of live nodes starting at around 250 iterations. Although the decline is slower than FQA, by 400 iterations, the number of live nodes reduces to zero. This pattern indicates that PSO-C is relatively more efficient in conserving node energy initially but struggles to maintain node survivability in the long term.

In contrast, the EOAMRCL protocol that implements GWO shows the best performance among the four protocols. The number of live nodes begins to decrease after about 100 iterations, similar to the other protocols, but the rate of decline is much slower. Even at 500 iterations, EOAMRCL manages to retain a significant number of live nodes, highlighting its superior capability in preserving node energy and maintaining network functionality over an extended period.

### 6.6. Discussion

The analyses of the seven figures collectively highlight the advantages and disadvantages of different Energy Optimization Approaches for WSNs. DWEHC consistently shows the poorest performance across all metrics, indicating inefficient energy management and high overhead costs. The rapid depletion of energy, early node deaths, high clustering overhead, and low packet reception rates all underscore DWEHC’s inadequacies in sustaining network performance. EEUC performs better but still experiences significant limitations in maintaining energy efficiency and network performance. While it shows improvements in energy consumption and node longevity compared with DWEHC, the protocol’s steady decline in live nodes and moderate clustering overhead reveal its insufficient optimization for extended network operations. CGA-GWO demonstrates notable improvements in energy management, node lifetimes, clustering overhead, and data transmission efficiency. The integration of the GWO algorithm allows for more efficient clustering and routing decisions, enhancing overall network performance. This is evidenced by the balanced energy consumption, delayed node deaths, and higher packet reception rates, indicating that CGA-GWO is effective in extending network life and reliability.

However, EOAMRCL outperforms all other protocols, showcasing the benefits of its cross-layer optimization approach. By integrating the MAC and network layers and leveraging GWO, EOAMRCL achieves superior energy efficiency, prolonged node lifetimes, reduced clustering overhead, and improved data transmission efficiency. The protocol maintains the highest average remaining energy, delays node deaths significantly, and ensures the highest percentage of live nodes across all iterations. Its low clustering overhead highlights its efficient resource management, while the high packet reception rates demonstrate its robustness in data transmission. The duty-cycle scheduling at the MAC layer, which alternates nodes between active and sleep modes, further conserves energy and extends the network’s operational life. This multifaceted optimization ensures that EOAMRCL not only addresses the limitations of single-layer protocols but also sets a new standard for energy-efficient management in WSNs. Overall, EOAMRCL proves to be the most effective solution for energy optimization in WSNs. Its comprehensive approach, which combines cross-layer optimization with advanced algorithms like GWO, highlights the importance of integrating multiple layers and techniques to achieve optimal network performance. The protocol’s ability to manage energy consumption efficiently, maintain network stability, and ensure reliable data transmission underscores its superiority. The findings from the analyses reinforce the critical role of advanced optimization methods in enhancing the sustainability and performance of Wireless Sensor Networks, making EOAMRCL a benchmark for future developments in this field.

In terms of comparing GWO with other optimization techniques, the analysis shows that while traditional techniques like FL and DV provide initial efficiency, they fail to maintain node survivability in the long term. Similarly, PSO shows better initial performance but ultimately faces significant node depletion issues. In contrast, GWO enables it to dynamically adjust and optimize clustering and routing strategies, leading to more efficient energy management and enhanced node longevity.

### 6.7. Strengths and Weaknesses of EOAMRCL

The strengths of EOAMRCL lie in its innovative integration of GWO and a cross-layer design, which collectively enhance the protocol’s energy efficiency and overall network performance. By leveraging GWO, EOAMRCL effectively balances exploration and exploitation phases, ensuring optimal cluster head selection. This results in a more efficient use of energy resources, as the most suitable nodes are chosen to manage communication within the network. Additionally, the cross-layer design of EOAMRCL improves energy efficiency by coordinating operations between the MAC and network layers. This coordination allows for a dynamic duty-cycle schedule, reducing unnecessary energy consumption by managing node activity more effectively. The protocol’s superior performance is demonstrated in its ability to extend network lifetime, maintain higher levels of residual energy, and achieve greater data throughput compared to other existing protocols.

However, the increased implementation complexity is a notable weakness of EOAMRCL. Integrating GWO and cross-layer interactions requires more sophisticated management and coordination, which can be challenging to implement and maintain. The computational overhead associated with optimizing CH selection and routing paths is also higher, potentially limiting the protocol’s applicability in resource-constrained environments. Additionally, while EOAMRCL shows excellent performance in static network scenarios, it may face challenges in highly dynamic environments where network topology and node energy levels change frequently. Adapting the protocol to such conditions could require further refinement and additional mechanisms to ensure scalability and adaptability.

## 7. Conclusions

This paper introduces a novel multi-layer protocol for energy-efficient management in WSNs, emphasizing the interaction between the MAC and network layers to minimize unnecessary energy consumption. We developed a robust objective function to identify the optimal cluster group and the best CHs during the formation phase while our routing protocol selects the most energy-efficient route for data delivery based on transmission power. Our new data transmission strategy for both intra-cluster and inter-cluster communication effectively addresses excessive energy consumption in the routing process. Each node schedules active and sleep modes using allotted NAV time slots, allowing the MAC layer to generate a duty-cycle schedule through cross-layer routing information. Additionally, the integration of a modified CSMA/CA mechanism with sleep/activate mode enhances the protocol’s energy efficiency by managing node activity more effectively. Simulation results demonstrate that EOAMRCL outperforms EEUC, CGA-GWO, and DWEHC protocols in terms of overall network remaining energy, number of dead nodes, total data received at the BS, and network lifetime. The superior performance is due to the efficient multi-layer approach, which overcomes the limitations of single-layer protocols. The innovative EOAMRCL protocol significantly enhances the energy efficiency and longevity of WSNs by leveraging cross-layer interactions. By integrating the MAC and network layers, our protocol ensures more precise and effective energy management, leading to a notable reduction in energy wastage. This multi-layer synergy is crucial for maintaining the balance between energy consumption and network performance, particularly in complex WSN environments. The use of NAV time slots for scheduling active and sleep modes further optimizes energy usage, enabling nodes to conserve energy without compromising data transmission reliability. Our extensive simulation results highlight EOAMRCL’s ability to maintain higher residual energy levels, fewer dead nodes, and greater data throughput at the base station compared with the other protocols. This is attributed to the protocol’s strategic approach to clustering and routing, which dynamically adjusts to the network’s energy states and communication demands. The robust objective function plays a vital role in selecting the most suitable CHs, ensuring that energy resources are utilized efficiently and effectively throughout the network’s operation.

Future work will explore the applicability of EOAMRCL in mobile sensor networks, where node mobility introduces additional challenges to energy management and network stability. Incorporating mobility into our protocol will require further refinement of the objective function to account for dynamic changes in node positions and energy levels. Additionally, we plan to evaluate the impact of incorporating more parameters into the objective function, such as node density and traffic load, to enhance its robustness and adaptability. We also intend to experiment with varying the weights assigned to different parameters within the objective function. This will help us understand how different prioritizations can affect the overall performance of the protocol, allowing for more tailored and context-specific implementations.

## Figures and Tables

**Figure 1 sensors-24-05234-f001:**
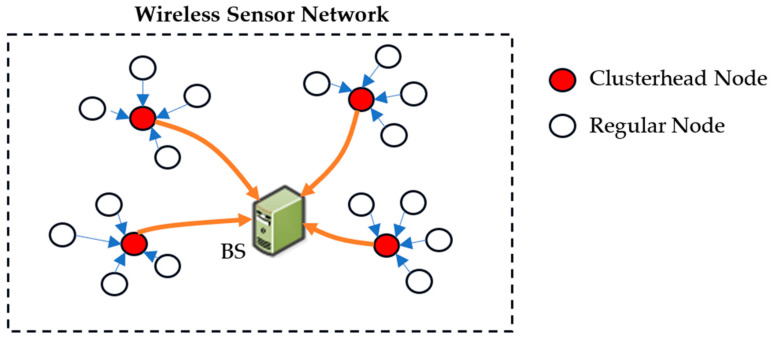
Clustered network topology.

**Figure 2 sensors-24-05234-f002:**
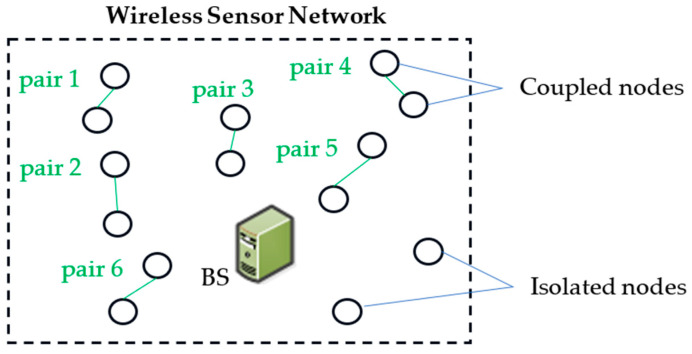
Node paring.

**Figure 3 sensors-24-05234-f003:**
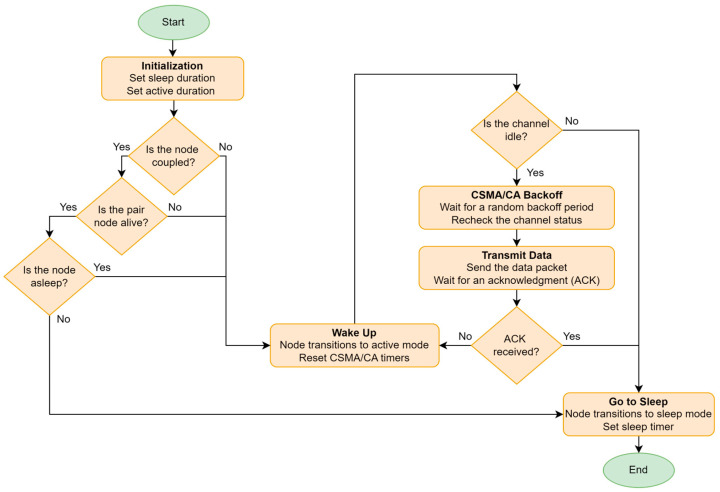
Integration of sleep mode in CSMA/CA mechanism.

**Figure 4 sensors-24-05234-f004:**
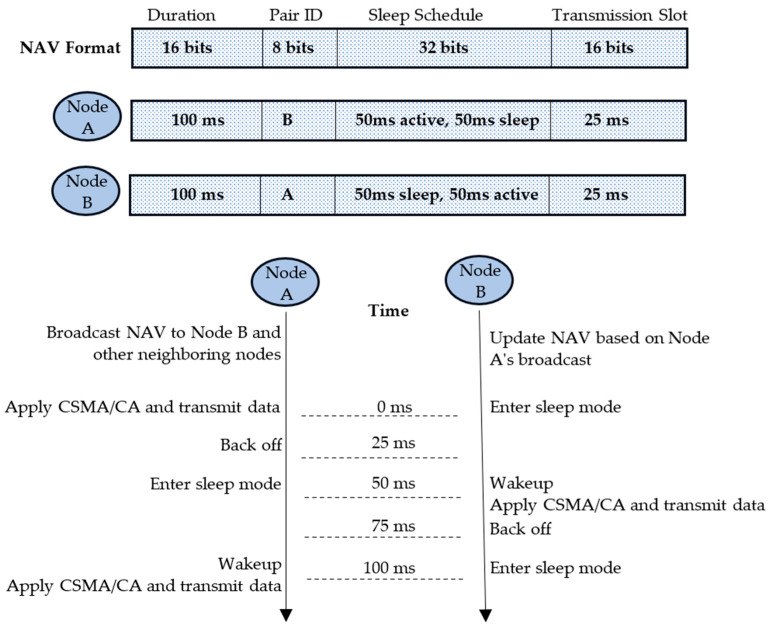
Integration of sleep mode in NAV.

**Figure 5 sensors-24-05234-f005:**
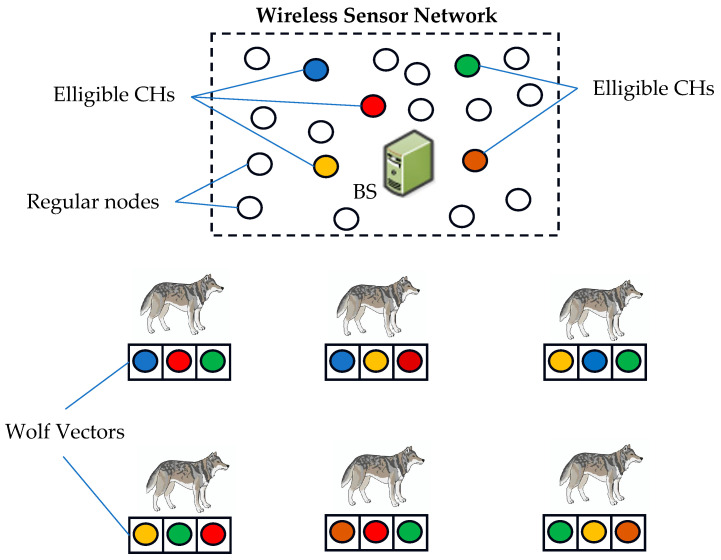
Construction of wolf vectors from eligible CHs.

**Figure 6 sensors-24-05234-f006:**
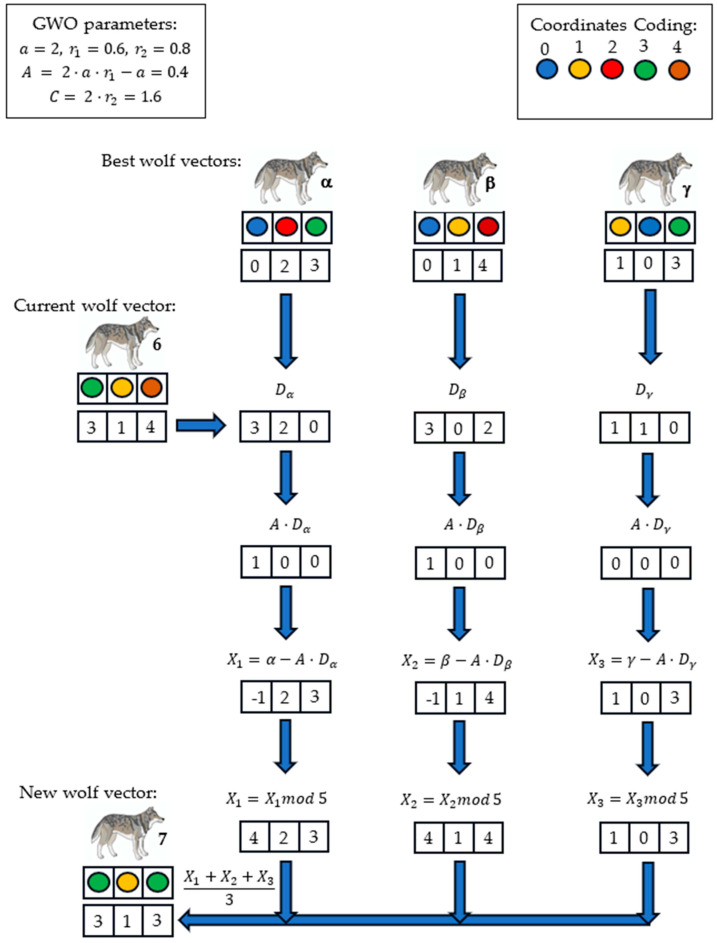
Example of wolf vectors update.

**Figure 7 sensors-24-05234-f007:**
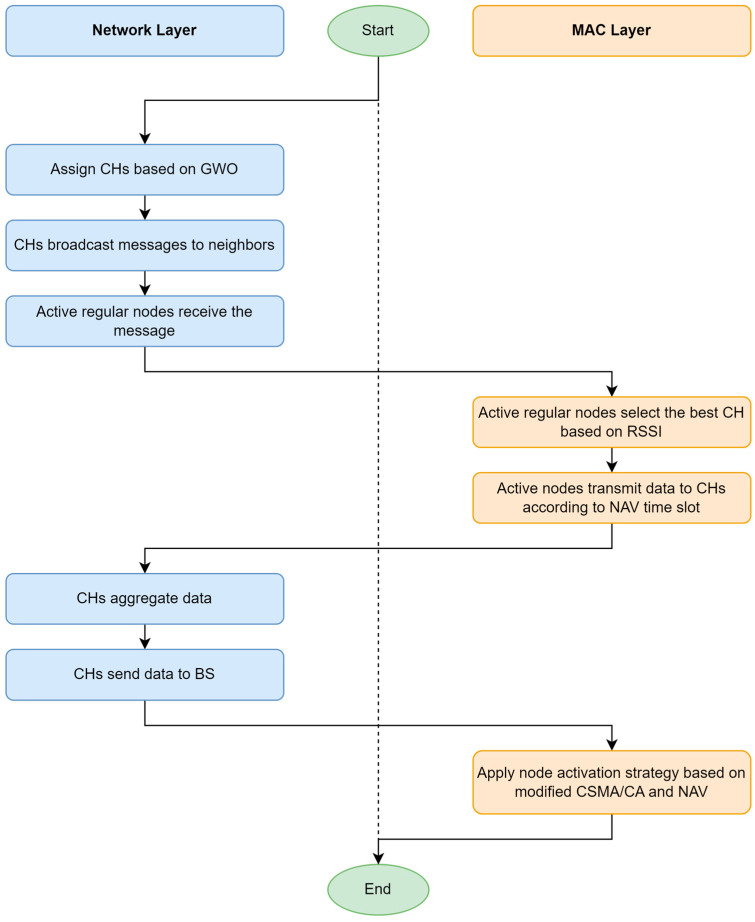
Cross-layer transmission phase.

**Figure 8 sensors-24-05234-f008:**
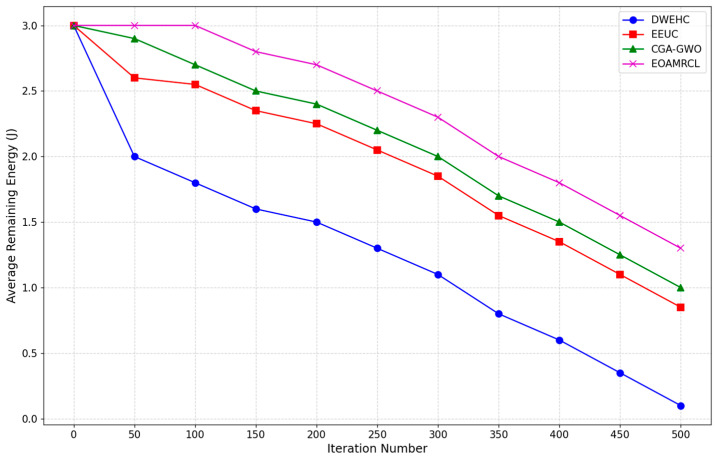
Comparison of average residual energy.

**Figure 9 sensors-24-05234-f009:**
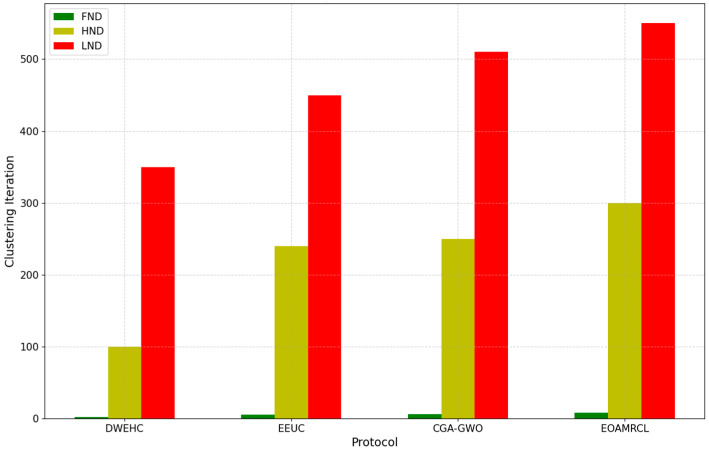
Comparison of FND, HND, and LND.

**Figure 10 sensors-24-05234-f010:**
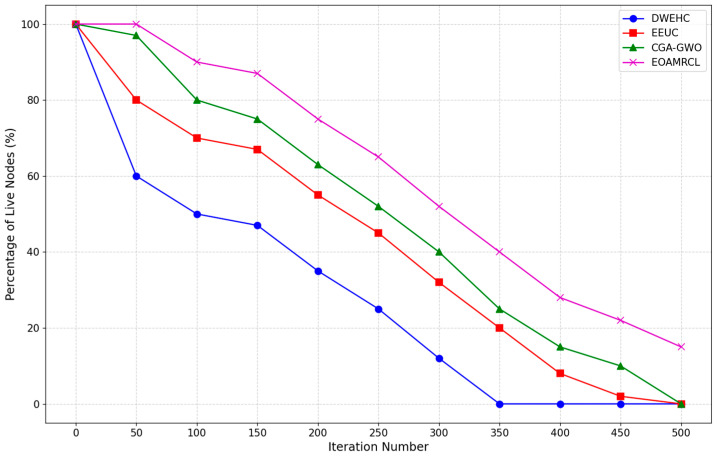
Comparison of live node percentage.

**Figure 11 sensors-24-05234-f011:**
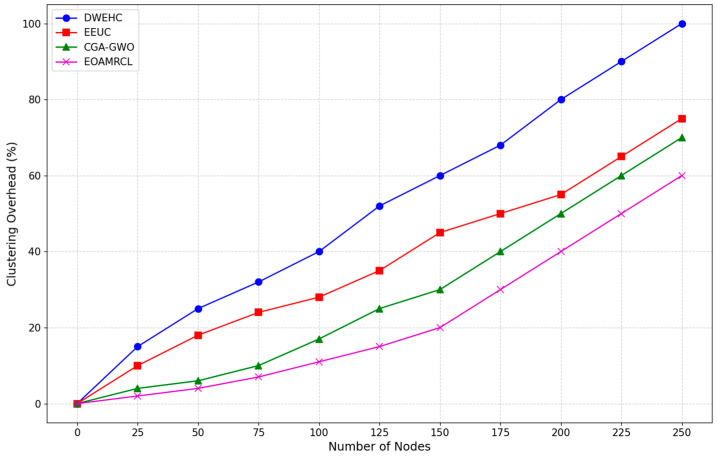
Comparison of clustering overhead.

**Figure 12 sensors-24-05234-f012:**
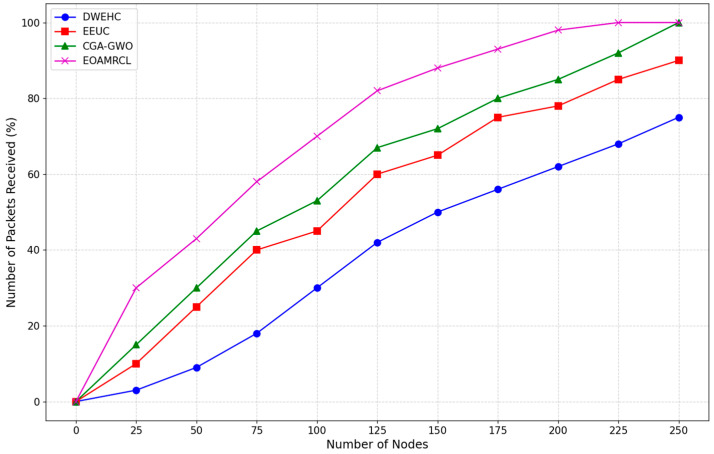
Comparison of network throughput.

**Figure 13 sensors-24-05234-f013:**
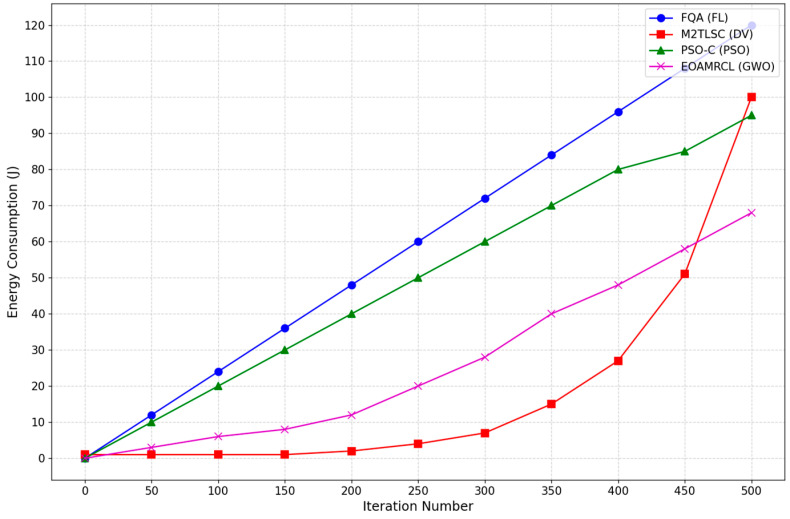
Comparison of energy consumption of different optimization techniques.

**Figure 14 sensors-24-05234-f014:**
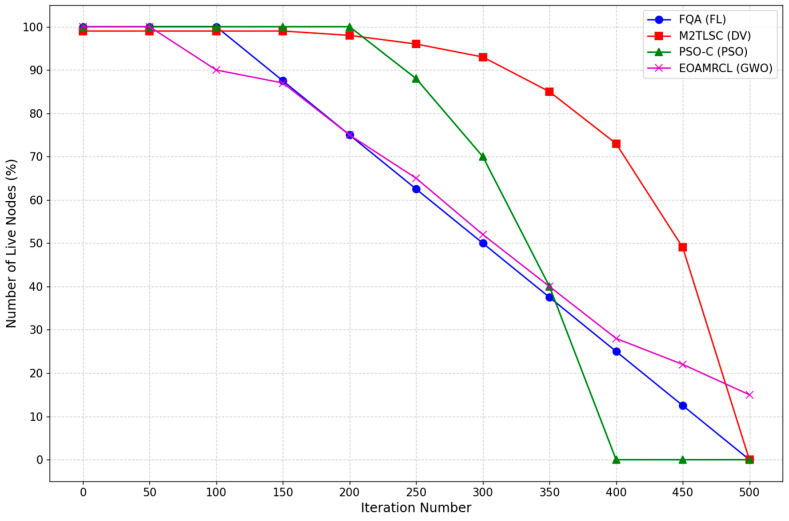
Comparison of live nodes of different optimization techniques.

**Table 1 sensors-24-05234-t001:** List of acronyms.

Acronym	Description
ACO	Ant Colony Optimization
AODV	Ad hoc On-demand Distance Vector
ARE	Average Residual Energy
AR-SC	Adjustable Range Set Covers
BACA	Binary Ant Colony Algorithm
BS	Base Station
BSTS	Bulk Service a Time Scheme
CCA	Clear Channel Assessment
CCBE	Cross-layer Cluster-Cased Energy-efficient
CEE	Cross-layer Energy Efficiency
CGA	Chaotic Genetic Algorithm
CH	Clusterhead
CL-MAC	Cross-Layer MAC
CREC	Cross-layer, Reliable, and Efficient Communication protocol
CSMA/CA	Carrier Sense Multiple Access with Collision Avoidance
DLC	Data Link Control
DV	Distance Vector
DWEHC	Distributed Weight-Based Energy-Efficient Hierarchical Clustering
EAP-CMAC	Energy Aware Physical-layer Network Cooperative MAC
ECC	Error Correction Codes
ECCA	Enhanced Clear Channel Assessment
EEUC	Energy-Efficient Unequal Clustering
EOAMRCL	Energy Optimization Approach based on MAC/Routing Cross-Layer
EQPD-MAC	Energy-aware QoS MAC protocol based on Prioritized Data and Multi-hop routing
FL	Fuzzy Logic
FIS	Fuzzy Inference System
FND, HND, LND	First Node Dead, Half of Nodes Dead, Last Node Dead
FQA	Fuzzy Logic with a Quantum Annealing Algorithm
GCRAD	Cross-layer Routing for Disaster
GCWGC	Greedy Coverage Weighted Communication
GWO	Grey Wolf Optimization
HC	Hill Climbing
HEED	Hybrid Energy-Efficient Distributed Clustering
IoT	Internet of Things
IP	Internet Protocol
LEACH	Low-Energy Adaptive Clustering Hierarchy
MAC	Medium Access Control
NAV	Network Allocation Vector
NS2	Network Simulator version 2
OCCH	Optimized Connected Coverage Heuristic
OSI	Open Systems Interconnection
OSTS	One Service a Time Scheme
OTTC	Overlapping Target and Connected Coverage
PHY	Physical Layer
PNC	Physical Layer Network Coding
QoS	Quality of Service
RSSI	Received Signal Strength Indication
SA	Simulated Annealing
TCP	Transmission Control Protocol
TDMA	Time-Division Multiple Access
TSLC	Topological Structure by Layered Configurations
WSN	Wireless Sensor Network

**Table 2 sensors-24-05234-t002:** Comparison of different MAC-Network routing techniques.

Protocol/ Technique	Involved OSI Layers	MAC Technique	Parameters Used	Routing	Scalability	Key Findings
EQPD-MAC [22]	Network MAC	TDMA	Residual Energy, Packet Priority, Multi-Hop Path	Multi Hop	High	Combines prioritized data handling with multi-hop routing for efficient energy usage
FQA [23]	Network MAC	TDMA	Residual Energy, Neighbors, Distance to BS, Node Centrality	Multi Hop	High	Combines fuzzy logic for CH selection with quantum annealing for optimal routing
SA, ECC [24]	Physical Data Link	TDMA	Coverage, Connectivity	Multi Hop	Medium	Lower power consumption and better network coverage compared to heuristics
EAP-CMAC [25]	Physical Data Link	CSMA/CA	Quality of Connection, Destination Queue	Multi Hop	Medium	Improved network lifespan and reduced power dissipation
GCRAD [26]	Data Link Network	ALOHA	Number of Relays, Node Queue State, Distance to BS	Multi Hop	High	Effective for disaster relief with reduced latency and power usage
ARSC, OCCH, CWGC, OTTC [27]	Physical Data Link Network	TDMA	Average Power Consumption, Network Lifespan	Single and Multi Hop	Medium	Insight into selecting appropriate algorithms based on specific network needs
CL-MAC [28]	Data Link Network	CSMA/CA	Network Conditions	Multi Hop	Medium	Enhanced data transmission efficiency and reduced energy consumption
MAC [29]	Data Link	TDMA	Idle Power, Duty Cycle	Multi Hop	Medium	Improved network lifespan and reduced idle power consumption
CREC [30]	Physical Data Link Network	CSMA/CA	Node Initiative, Congestion Management, Channel Effects	Multi Hop	High	Significant energy usage reduction and better network performance
TSLC [31]	Data Link Network	CSMA/CA	Node Status, Energy Consumption	Multi Hop	High	Enhanced energy conservation and prolonged network lifespan
CEE [32]	Data Link Network	CSMA/CA	Node Placement, Full-duplex Interfaces	Multi Hop	High	Effective for mobile networks with significant energy efficiency and performance improvements
CCBE [33]	Physical Data Link Network	TDMA	Distance to BS, Residual Energy, Slot Assignment	Multi Hop	High	Superior energy efficiency and network longevity compared to traditional clustering protocols
BACA, HC, SA [34]	Physical Data Link Network	TDMA	Sensor Placement, Sensing Coverage	Single Hop	Medium	Achieved high sensing coverage
CGA-GWO [35]	MAC Network	TDMA	Distance to BS, Residual Energy	Multi Hop	High	Combines CGA and GWO for efficient clustering and routing

**Table 3 sensors-24-05234-t003:** Comparison of different CSMA/CA-based routing protocols.

Protocol/ Technique	Involved OSI Layers	MAC Technique	Parameters Used	Physical Layer Features	Data Link Layer Features
Markov Model [36]	Physical MAC	CSMA/CA	Duty Cycle, Sleep Mode, Active/Sleep Transitions	Energy-efficient transmission, minimized idle listening	Duty cycle optimization, sleep mode transitions
OSTS, BSTS [38]	Physical MAC	CSMA/CA	Buffered Conditions, Channel Assessment, Sleep Scheduling	Optimized signal transmission, reduced interference	Buffer management, sleep scheduling
Enhanced CCA Mechanism [39]	Physical MAC	CSMA/CA	Signal Strength, Interference, Channel State	Improved channel sensing, interference handling	Enhanced CCA checks, adaptive strategies

**Table 4 sensors-24-05234-t004:** Simulation settings for network simulations.

Parameter	Value
Network Zone	100 × 100 m^2^
Number of Sensors (n)	50–250
BS Coordinates	(90,90)
$\mathrm{Clusterhead}\;\mathrm{Percentage}\;(P_{opt}$)	5%
Advanced Node Percentage (m)	20%
$\mathrm{Initial}\;\mathrm{Energy}\;(E_{0}$)	3 J/node
Additional Energy Factor (θ)	1
$\mathrm{Transmission}\;\mathrm{Energy}\;(E_{elec}$)	50 nJ/bit
Packet Size (L)	4000 bits
$\mathrm{Propagation}\;\mathrm{Energy}\;(\mathrm{fading}\;\mathrm{space}\;\varepsilon_{fs}$)	15 pJ/bit/m^2^
$\mathrm{Propagation}\;\mathrm{Energy}\;(\mathrm{multi}-\mathrm{path}\;\varepsilon_{mp}$)	0.0015 pJ/bit/m^4^
$\mathrm{Data}\;\mathrm{Aggregation}\;\mathrm{Energy}\;(E_{DA}$)	5 nJ/bit/signal
Node Pairing Distance	<2 m
$\mathrm{Fitness}\;\mathrm{Function}\;\mathrm{Weights}\;(w_{1},w_{2},w_{3}$)	0.45, 0.45, 0.1

## Data Availability

No new data were created, and all simulation results are presented in this paper.

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
