# Peer review of "Energy-Efficient Clustering in Wireless Sensor Networks Using Grey Wolf Optimization and Enhanced CSMA/CA"

_sensors, 2024, doi:10.3390/s24165234_

Round 1

Reviewer 1 Report

Comments and Suggestions for Authors

1. Can you explain why you chose Grey Wolf Optimization for your method in the Introduction?

2. The Related Work section is a bit lengthy. Could you condense it and compare your work with existing studies, noting the strengths and weaknesses?

3. It would be helpful to include an analysis of the algorithm's complexity.

4. Please provide a list of the simulation tools you used and their configurations.

5. You mentioned the concept of cross-layer design. Can you clarify how clustering impacts routing efficiency and how these two elements are connected?

Author Response

  1. Can you explain why you chose Grey Wolf Optimization for your method in the Introduction?

We appreciate the reviewer's insightful question regarding the choice of GWO for our method. We selected GWO due to several compelling reasons which we believe make it particularly well-suited for our application in WSNs. We have revised Section 4 to include this explanation, ensuring that our rationale for choosing GWO is clearly articulated.

  1. The Related Work section is a bit lengthy. Could you condense it and compare your work with existing studies, noting the strengths and weaknesses?

We appreciate the reviewer’s suggestion to condense the related work. We have revised section 2 and highlighted in section 6.7 the strengths and weaknesses of our approach.

  1. It would be helpful to include an analysis of the algorithm's complexity.

Thank you for your insightful comment and suggestion regarding the complexity analysis of our algorithm. We have provided a detailed examination in section 5.4.

  1. Please provide a list of the simulation tools you used and their configurations.

Thank you for this feedback. We provided a list of the simulation tools and environment in section 6.1.

  1. You mentioned the concept of cross-layer design. Can you clarify how clustering impacts routing efficiency and how these two elements are connected?

Thank you once again for your insightful comments. We added section 5.6 to clarify the interconnection between clustering and routing within our proposed cross-layer design.

Reviewer 2 Report

Comments and Suggestions for Authors

In this paper, we propose an energy optimization method called EOAMRCL that integrates the Gray Wolf Optimization (GWO) algorithm to enhance the performance of Wireless Sensor Networks (WSNs).EOAMRCL aims to increase the energy efficiency by optimizing the duty cycle scheduling, transmission power, and routing path selection to extend the network lifetime. The method employs a centralized strategy and hierarchical network architecture to determine the ideal cluster head (CH) using the GWO algorithm in the cluster formation phase, selects the path with the least energy consumption through routing protocols in the data transmission phase, and forms a duty cycle scheduling based on cross-layer routing information at the MAC layer. In addition, the method introduces an enhanced CSMA/CA mechanism to further minimize collisions, improve channel evaluation accuracy and reduce energy consumption.

Review comments:

1) The authors drew the flow of the Gray Wolf algorithm in detail, which is very interesting and vivid, but there are many intelligent optimization algorithms and many improvements for intelligent optimization algorithms, and the authors' reasons for choosing the GWO algorithm should be explained in detail.

2) The author's experimental simulation is more than adequate, but I suggest the author to compare other optimization algorithms in the experimental simulation part.

3) The analysis of the theoretical performance of the EOAMRCL method, such as convergence analysis and complexity analysis, can be added to enhance the theoretical depth of the paper.

4) I suggest the authors to add in the introduction to elaborate on other areas of the Gray Wolf optimization algorithm in WSN, which will enrich the introduction of the paper, for example:

1.Application of Gray Wolf Particle Filter Algorithm Based on Golden Section in Wireless Sensor Network Mobile Target Tracking

2.A UAV Path Planning Method in Three-Dimensional Space Based on a Hybrid Gray Wolf Optimization Algorithm

3.Estimation of Knee Joint Extension Force Using Mechanomyography Based on IGWO-SVR Algorithm

Overall, the article is well written, but I think some experiments need to be added.

Author Response

1) The authors drew the flow of the Gray Wolf algorithm in detail, which is very interesting and vivid, but there are many intelligent optimization algorithms and many improvements for intelligent optimization algorithms, and the authors' reasons for choosing the GWO algorithm should be explained in detail.

We appreciate the reviewer's insightful question regarding the choice of GWO for our method. We selected GWO due to several compelling reasons which we believe make it particularly well-suited for our application in WSNs. We have revised Section 4 to include this explanation, ensuring that our rationale for choosing GWO is clearly articulated.

2) The author's experimental simulation is more than adequate, but I suggest the author to compare other optimization algorithms in the experimental simulation part.

We appreciate your suggestion and we have considered incorporating comparisons with other optimization algorithms like fuzzy logic, distance vector and particle swarm optimization (Figures 13 and 14).

3) The analysis of the theoretical performance of the EOAMRCL method, such as convergence analysis and complexity analysis, can be added to enhance the theoretical depth of the paper.

Thank you for your insightful comment and suggestion regarding the complexity analysis of our algorithm. We have provided a detailed examination in section 5.4.

4) I suggest the authors to add in the introduction to elaborate on other areas of the Gray Wolf optimization algorithm in WSN, which will enrich the introduction of the paper, for example:

1.Application of Gray Wolf Particle Filter Algorithm Based on Golden Section in Wireless Sensor Network Mobile Target Tracking

2.A UAV Path Planning Method in Three-Dimensional Space Based on a Hybrid Gray Wolf Optimization Algorithm

3.Estimation of Knee Joint Extension Force Using Mechanomyography Based on IGWO-SVR Algorithm

We appreciate the suggestion to elaborate on the applications of the GWO algorithm in WSNs and include additional references to enrich the context and underscore the versatility of GWO in various domains. We have added a new paragraph in the introduction to address this comment (just before paper contribution). Additionally, we have incorporated the three suggested articles into the references section (ref. 19, 20, and 21), in addition to another article that implements PSO in WSN (ref. 47).

Round 2

Reviewer 1 Report

Comments and Suggestions for Authors

The whole structure of the article and the specific research content have been revised as required, and there are no further comments.

TRANSLATE with x English
Arabic Hebrew Polish
Bulgarian Hindi Portuguese
Catalan Hmong Daw Romanian
Chinese Simplified Hungarian Russian
Chinese Traditional Indonesian Slovak
Czech Italian Slovenian
Danish Japanese Spanish
Dutch Klingon Swedish
English Korean Thai
Estonian Latvian Turkish
Finnish Lithuanian Ukrainian
French Malay Urdu
German Maltese Vietnamese
Greek Norwegian Welsh
Haitian Creole Persian  
TRANSLATE with COPY THE URL BELOW Back EMBED THE SNIPPET BELOW IN YOUR SITE Enable collaborative features and customize widget: Bing Webmaster Portal Back Comments on the Quality of English Language

The overall language quality of the article is stable, the language description is objective, the nuances are slightly grammatically deficient, the overall language quality is feasible, there are no more comments.

TRANSLATE with x English
Arabic Hebrew Polish
Bulgarian Hindi Portuguese
Catalan Hmong Daw Romanian
Chinese Simplified Hungarian Russian
Chinese Traditional Indonesian Slovak
Czech Italian Slovenian
Danish Japanese Spanish
Dutch Klingon Swedish
English Korean Thai
Estonian Latvian Turkish
Finnish Lithuanian Ukrainian
French Malay Urdu
German Maltese Vietnamese
Greek Norwegian Welsh
Haitian Creole Persian  
TRANSLATE with COPY THE URL BELOW Back EMBED THE SNIPPET BELOW IN YOUR SITE Enable collaborative features and customize widget: Bing Webmaster Portal Back

Reviewer 2 Report

Comments and Suggestions for Authors

 Accept in present form